# The Use of Concentrates Rich in Orange By-Products in Goat Feed and Its Effects on Physico-Chemical, Textural, Fatty Acids, Volatile Compounds and Sensory Characteristics of the Meat of Suckling Kids

**DOI:** 10.3390/ani10050766

**Published:** 2020-04-28

**Authors:** José Luis Guzmán, Manuel Delgado-Pertíñez, María José Beriáin, Rafael Pino, Luis Ángel Zarazaga, Alberto Horcada

**Affiliations:** 1Departamento de Ciencias Agroforestales, ETSI, Universidad de Huelva, “Campus de Excelencia Internacional Agroalimentario, ceiA3”, Campus de la Rábida, 21819 Palos de la Frontera, Huelva, Spain; 2Departamento de Ciencias Agroforestales, ETSIA, Universidad de Sevilla, 41013 Sevilla, Spain; 3Research Institute for Innovation & Sustainable Development in Food Chain (ISFOOD), Universidad Pública de Navarra, 31006 Pamplona, Spain; 4Departamento de Estadística e Investigación Operativa, Facultad de Matemáticas, Universidad de Sevilla, 41012 Sevilla, Spain

**Keywords:** by-products, feeding sources, goats, orange pulp

## Abstract

**Simple Summary:**

Spain is a major global producer of both goats and citrus fruits on the world. Using by-products of the orange industry for feeding ruminants has environmental advantages. In this work, we analysed how replacing cereal concentrates with dehydrated orange pulp (DOP) in the diet of mother goats affects the meat quality of suckling kids. We evaluated the following characteristics of the meat of suckling kids of the dairy Payoya breed: chemical composition; texture; water holding capacity; colour; saturated, monounsaturated, and polyunsaturated fatty acids (SFA, MUFA, and PUFA, respectively); volatile compounds; and sensorial appraisal. The inclusion of DOP in goat feed did not affect the proximal composition, texture, colour, or juiciness of the kids’ meat. However, the inclusion of DOP improved the indices of the nutritional value of the meat for human health (thrombogenicity index, PUFA/SFA ratio, and n-6/n-3 ratio). The inclusion of DOP in goat feed reduced MUFA content in the kids’ meat. An increase in aromatic compounds, including ethyl furan, dimethyl disulphide, and heptane, was observed in the grilled meat of kids from goats that were fed DOP. The use of DOP in goat feed improved consumers’ sensory appreciation of the suckling kids’ meat.

**Abstract:**

We analysed how replacing cereal concentrates with dehydrated orange pulp (DOP) in the diet of mother goats affects the meat quality of suckling kids. Three experimental diets for mother goats were designed. The DOP-0 diet contained commercial concentrates and alfalfa hay. In the DOP-40 and DOP-80 diets, 40% and 80% (respectively) of the cereal in the concentrate was replaced with pellets of DOP (the alfalfa hay component was unchanged). We evaluated the chemical composition, texture, water holding capacity, colour, fatty acids (FAs) profile, volatile compounds, and sensorial appraisal of the meat from 30 male suckling kids (cold carcass weight 4.74 kg, 4.82 kg, and 4.65 kg for DOP-0, DOP-40, and DOP-80, respectively) of the Payoya breed (n = 10 for each diet). Meat from kids in the DOP-40 and DOP-80 groups exhibited characteristics favourable for human health, including the meat’s thrombogenicity index, PUFA/SFA ratio (0.60 index), and n-6 PUFA/n-3 PUFA ratio (approximately 7.50). The meat also exhibited reduced MUFA content (around 460 mg/100 g fresh meat). An increase in ethyl furan, dimethyl disulphide and heptane was observed in grilled meat from goats that were fed using DOP. The inclusion of DOP in goat feed improved consumers’ sensory appreciation of the kid’s meat.

## 1. Introduction

Spain is a major producer of both goats and dehydrated orange pulp (DOP). In 2017, Spain had the second-highest number of goats of any European country (3.06 × 10^6^) and was the top producer of oranges in Europe (3.35 × 10^6^ t) [1]. Goats are mainly raised for milk production for the cheese industry. However, goat farms also produce suckling kid goats, which are exclusively fed with mother’s milk for one month until they are slaughtered with a live weight of 8–10 kg [2].

Approximately 75% of Spanish orange production is devoted to fresh consumption and the remainder is for industrial production of orange juice [3]. The orange juice industry produces organic waste that can be used for other purposes. More specifically, DOP is a by-product of fruit juice extraction and consists of the dried residue of orange peels, pulp, and seeds. The processing of orange juice can generate waste of up to 15% *w/w* with respect to input, so it is important to consider options for its reuse. One possibility is to include orange waste in animal feed in the form of dried pellets, especially as a high-energy feed for ruminants to support growth and lactation. Technological advances have been made in the manufacture of animal feed, and the manufacture of dried feed pellets using orange by-products is now common. This represents a qualitative advantage to using orange by-products in animal feed.

Previous studies have focused on the effects of citrus pulp on animal growth and nutrition [4], but little is known about the effects of orange pulp on meat quality. Several studies have investigated the use of DOP by-product feed for sheep and have assessed the feed’s effect on the meat quality of lamb. Some of these studies [5] have demonstrated the advantages of replacing the cereals in concentrate-based sheep diets with dried DOP as a feasible strategy to naturally improve the meat’s oxidative stability. This is possible because using DOP by-products in sheep diets generates more moderate fermentations that can alter the profile of volatile fatty acids, basic milk compounds, and precursors of fatty acids (FAs) in meat [6]. Caparra et al. [7] found that the inclusion of dried DOP in lamb diets produced meat with lower redness values compared with the meat from lambs raised on a grain-based diet.

In Spain, the Payoya breed is being re-oriented to an intensive milk production system. Therefore, goat’s farmers require feeding alternatives to achieve greater milk yields. The use of local alternative feeds can offer economic advantages by reducing feeding costs and mitigating the adverse socio-environmental impacts that would otherwise arise from the disposal of agri-industrial by-products [8]. The objective of the present study is to assess the effect of replacing cereal concentrates in goat feed with two levels of dehydrated orange pulp on the meat quality of the goats’ suckling kids.

## 2. Materials and Methods

### 2.1. Animal Management and Dietary Treatments

Thirty male Payoya suckling kid goats were used for the feeding trial on an experimental farm at Huelva University (Spain). Previously, 44 first-kidding goats were distributed into three experimental groups according to isoenergetic and isoproteic dietary treatments. The groups were balanced according to their live weight and body condition. The three diets of feed were as follows: DOP-0 (n = 14) composed of commercial concentrates with alfalfa hay, DOP-40 (n = 16) composed of commercial concentrates and alfalfa hay with 40% of concentrate cereals replaced with DOP pellets, and DOP-80 (n = 14) composed of commercial concentrates and alfalfa hay with 80% of concentrate cereals replaced with DOP pellets (Table 1). The mean live weights of the goats in each group were 37.0 ± 1.26, 38.0 ± 1.33 and 37.5 ± 2.12 kg for the DOP-0, DOP-40, and DOP-80 groups, respectively. The groups’ mean body condition values were 2.76 ± 0.09, 2.51 ± 0.06, and 2.55 ± 0.08 for the DOP-0, DOP-40 and DOP-80 groups, respectively. From the beginning of the third month of gestation, DOP was gradually introduced into the diet of the animals in the DOP-40 and DOP-80 groups, and before application the experimental diets, visual observation of feed intake indicated that goats consumed all DOP pellets offered. Thus, changes in feed composition did not influence the palatability of the diets. During the last month of pregnancy, each group of animals received its corresponding experimental prepartum diet (on average, goats were fed 0.45 kg/d of alfalfa hay and 0.7 kg/d of concentrate per animal), with a 60:40 concentrate-to-forage ratio. After parturition and during the suckling period, the animals were given the experimental diets adapted to early lactation (Table 1; on average, goats were fed 0.4 kg/d of alfalfa hay and 1.67 kg/d of concentrate per animal), which consisted of a concentrate-to-forage ratio of 80:20. Although the individual intake was not measured, the average intake per animal obtained indirectly through the consumption of the group shows similar values in the different dietary treatments (DM was 1.85, 1.82 and 1.80 kg/d, protein digestible in the intestine was 0.19, 0.20 and 0.20 kg/d, and gross energy was 8.42, 8.10 and 8.00 Mcal/d in the DOP-0, DOP-40 and DOP-80 groups, respectively).

After parturition, kids remained with their dams from birth to weaning with free access to goat milk all day. All the kids were only fed with maternal milk until slaughtered. Ten male suckling kid goats were selected randomly from each experimental group and slaughtered at an average weight of 8.90 kg (±0.17) and 32 (±1.5) days old.

### 2.2. Sampling and Chemical Analysis of Diet and Mother’s Milk

For each diet, samples of hay (2.5 kg) and concentrate (2 kg) were prepared for analysis by mixing equal amounts of three subsamples collected during the suckling period (at 10, 20 and 30 days after birth). The samples were stored at −4 °C. Samples of experimental diets were ground to pass a 1 mm screen in a cyclone mill (Foss) before analysis. The diet components were determined using AOAC methods, as follows: dry matter (method 934.01), ash (method 942.05), ether extract (method 920.39), N (method 984.13), and crude fibre (method 978.10) [9]. The N values were determined by the Kjeldahl procedure in a 2200 Kjeltec auto-distillation unit (FossTecator, Höganäs, Sweden) and converted to crude protein by multiplying by a factor of 6.25. Fat content was measured by extraction with petroleum ether (boiling point, 40–60 °C) using a Soxtec System HT 1043 Extraction Unit (Foss Tecator, Sweden). The crude fibre was analysed using a Fibertec 2010 Hot Extraction Unit (Foss Tecator, Sweden). The forage units for lactation (UFL) and protein digestible in the small intestine (PDI) were calculated using the Feed Ration Balancer (Format Solutions) software tool, version 2.0 (2017; Cargill, Inc., Minneapolis, MN, USA; www.formatsolutions.com).

Milk was sampled twice during the suckling period (third and fourth week post-partum) and representative samples were taken from each dam. Samples were deposited in iceboxes, sent to the laboratory, and stored at −20 °C until they were analysed. The traits of milk samples (dry matter, protein, and fat content) taken during the suckling period were estimated by near-infrared spectroscopy with a Foss NIRSystems 6500 SY-I monochromator (FOSS-NIR Systems Inc., Silver Spring, MD, USA) according to Delgado-Pertíñez et al. [10]. The metabolisable energy (ME) concentration was estimated according to Nsahlai et al. [11].

### 2.3. Slaughter Procedures and Muscle Sampling

The suckling kids were slaughtered using standard commercial procedures according to the guidelines of Council Regulation (EC) Nº 1099/2009 [12] on the protection of animals at the time of killing. After refrigeration for 24 h at 4 °C, the carcasses were weighed and cut into halves along the midline. The pH values in the *longissimus lumborum* (LL) muscle at the 4th and 5th lumbar vertebra were measured using a pH meter (Crison 507; Crison Instruments, Barcelona, Spain) equipped with a penetrating glass electrode. The meat colour was measured at the same time and location as the pH readings. The colour was determined as the mean of three measurements taken directly on the LL surface after removing any other connective tissue with a scalpel. The colour measurement process included one hour of blooming using a Konica Minolta CM colorimeter (Konica Minolta Inc., Tokyo, Japan) in the CIELAB colour space [13] with a standard illuminant D65, an observation angle of 10°, and zero and white calibration. The lightness (L*), redness (a*) and yellowness (b*) of the meat were recorded. The hue angle (H°) was calculated as tan^−1^(b*/a*) and the chroma (C*) as [(a*2 + b*2)^1/2^].

The left half of the carcass was transported to the meat quality laboratory of Huelva University and the shoulder was dissected to obtain the *triceps brachii* (TB) muscle. The chemical composition of the TB muscle was determined by duplicate according to standard AOAC methods [9] and expressed on a wet basis: the moisture content by method 24,003, the intramuscular fat content by method 13,032, the nitrogen content by method 2057, and the ash content by method 14,066.

After removing the rib cut, the *longissimus thoracis* (LT) muscle was extracted and two portions of muscle (5 g each) were used to determine the water-holding capacity (WHC) of meat according to Guzmán et al. [14]. The WHC was expressed as the percentage of expelled juice after compression. The rest of the muscle was vacuum packed and frozen at −20 °C until further analysis.

After thawing the LT muscle under chilled conditions (4 °C) for 24 h, the cooking loss, Warner–Brätzler shear force (WBSF), and haem pigment content were analysed [15]. The haem pigment concentration was calculated using a physical-chemical method expressed as mg myoglobin/g fresh meat. To determine the cooking loss, a portion of caudal slices of vacuum-packed LT was weighed and heated in a 75 °C water bath to an internal temperature of 70 °C and monitored with a Jenway thermocouple equipped with a probe (Hanna Instruments HI 8757). The cooking loss was calculated as the difference between the weight of the loin portion before packaging and its weight after cooking, expressed as a percentage [16]. The Cooking loss test samples were then used to determine the WBSF using a Stevens QTS 25 texture analyser equipped with a Warner–Brätzler device [14]. The slices were then cut into samples with a cross-section of 1 cm^2^ parallel to the muscle fibres. The maximum shear force (kg/cm^2^) was assessed parallel to the muscle fibres in at least three subsamples of heated meat pieces.

### 2.4. Fatty Acid Analysis

The FA profile of the feed and milk was determined via gas chromatography according to conditions reported by Gutiérrez-Peña et al. [17]. FA methyl esters were separated and quantified using a gas chromatograph (Agilent 6890N Network GS System, Agilent, Santa Clara, CA, USA) equipped with a flame-ionisation detector (FID), an HP 7683 automatic sample injector, and an HP-88 J&W fused silica capillary column (100 m, 0.25 mm i.d., 0.2-µm film thickness; Agilent Technologies Spain, S.L., Madrid, Spain). The FA profile of intramuscular fat from LT muscle was analysed according to Horcada et al. [18] using a chromatograph (GC, Agilent 6890N, Inc., Santa Clara, CA, USA) equipped with a FID and fitted with a BPX-70 capillary column (120 m, 0.25 mm i.d., 0.2 μm film thickness, SGE, Postnova Analytics Inc., Salt Lake City, UT 84102, USA). The chromatographic conditions were as follows: the initial column temperature was 100 °C, programmed to increase at a rate of 3 °C /min up to 158 °C and then at 1.5 °C/min up to 190 °C, maintaining this temperature for 15 min, then at 2 °C /min up to 200 °C and at 10 °C /min up to a final temperature of 240 °C, which was maintained for 10 min. The injection and detector temperatures were maintained at 300 °C and 320 °C, respectively. Hydrogen was used as a carrier gas at a flow rate of 2.7 mL/min. Fatty acid methyl esters in hexane (1 µL) were automatically injected into the column with a split ratio of 17.7:1. Individual FAs were identified using standards (Sigma Chemical Co. Ltd., Poole, UK), expressed as mg FA/100 mg fresh meat, and grouped as follows: saturated (SFA), monounsaturated (MUFA) and polyunsaturated (PUFA). The atherogenicity index (AI) [C12:0 + 4 × 14:0 + C16:0]/[MUFA + PUFA] and the thrombogenicity index (TI) [C14:0 + C16:0 + C18:0]/[0.5 × MUFA + 0.5 × n-6 PUFA + 3 × n-3 PUFA + (n-3 PUFA/n-6 PUFA)] were calculated according to Ulbricht and Southgate [19].

### 2.5. Volatile Compound Analysis

Slices of the LT muscle were stored overnight at 4 °C before analysis and were cooked separately at 200 °C under a mixed closed griddle (Jatta Electro, GR266 1000W, Abadiano, Vizcaya, Spain). The grill was switched on for 15 min before the sample was grilled. Each sample was placed in the middle of the grilling tray to ensure uniform grilling and was cooked to a core temperature of 70 °C [20]. Directly after cooking, the meat and all the fat released during cooking were chopped finely in an electric bowl chopper (Janke and Kunkel A-10, IKA Labortechnik, D-79219 Staufen, Germany). Approximately 10 g of the sample was analysed. Volatile compounds were extracted via the dynamic headspace technique [21]. Immediately after chopping, the samples were placed in a headspace vial (Tekmar, 100 mL). The vial was attached to a purge-and-trap sample concentrator (model 4460A, OI Analytical, College Station, TX, USA) and was heated in an external heater set to 70 °C. Volatile compounds were separated in an HP-6890 gas chromatograph (Hewlett-Packard, Madrid, Spain) connected to an HP-5973 quadrupole mass spectrometer (Hewlett-Packard, Madrid, Spain) using an HP-5 capillary column (5% phenyl methyl siloxane; 50 m × 320 μm i.d. × 1.05 μm film thickness) (Hewlett-Packard, Madrid, Spain) with helium as the carrier gas [21]. A series of n-alkanes (C5–C18¸ HP5080-8768) in diethyl ether was analysed under the same conditions to obtain retention index (RI) values for each volatile compound according to the following equation [22]: RI = 100*z* + 100 (RT*i* − RT*z*) (RT*z*+1 − RT*z*) where RI is the retention index of the unknown peak, RT*i* is the retention time for the unknown peak, RT*z* and RT*z*+1 are the retention times for the n-alkanes that bracket the unknown peak, and *z* is the number of carbon atoms in the n-alkane standard that elute just before the unknown peak.

Compounds were identified by comparing their mass spectra with those contained in the Wiley 275 Library Mass Spectral Database or in previously published literature. Wherever possible, compound identities were confirmed by comparing their RI values with either authentic standards or published values. Retention indices (IDB-5) were calculated using a modified version of the method for temperature-programmed gas chromatography [23]. The peak area of the volatile compounds was integrated from specific ions for each molecule to avoid overlapping between compounds. The integrations were performed using the Chemstation software tool (Hewlett-Packard, Palo Alto, CA, USA).

### 2.6. Sensorial Evaluation

For the sensorial analysis, the left legs of suckling kid goats were used once thawed under chilled conditions (4 °C) for 24 h. The samples were evaluated by 30 untrained panellists consisting of members over the age of 18 years from 10 families (three panellists from each family). Each family received one leg from each diet treatment (DOP-0, DOP-40, and DOP-80) and each of the three family members who participated in the study evaluated the leg from each treatment according to a repeated measures design (n = 90 ratings). According to the recommendations for cooking samples, the legs were prepared with olive oil and salt in a standard oven set to 220 °C for 60 min. The sensorial analysis was carried out according to the method described by Martínez-Cerezo et al. [24]. The following attributes were rated on a 10-point scale: tenderness, juiciness, flavour quality, and overall appraisal. The tenderness was rated from 1 = very tough to 10 = very tender; juiciness from 1 = not at all juicy to 10 = very juicy; and flavour quality from 1 = very bad to 10 = very good. The overall appraisal was rated from 1 = very bad to 10 = very good.

### 2.7. Statistical Analysis

Each parameter of the meat quality was analysed with ANOVA procedures. Normality and homoscedasticity assumptions were rejected for many variables according to the Shapiro-Wilk and Levene tests. Therefore, we chose to use a permutation test using the F-statistic computed in the parametric ANOVA. Under the null hypothesis, in which it is assumed that the “goat feed” factor had no effect, it is equally likely that any of the individual factor labels could be associated with any of the samples. Therefore, if the groups do not differ, another possible value of the test statistic under the null hypothesis may be obtained by randomly shuffling the group labels. In the permutation test, this random shuffling is repeated a large number of times (B) to obtain B values for the test statistic. A *p* is computed as the proportion of the B values that are greater than or equal to the observed F statistics in the data set. In this study, 999 permutations were performed (B = 999). This procedure was carried out with the adonis function in the R-vegan package [25].

The statistical model used in the ANOVA included the fixed effects of the feeding regime and the random effect of the individual. The following model was used:Y_ik_ = µ + FR_i_ + e_ik_
where Y_ik_ is pH, moisture, protein, fat, ash content, shear force, WHC, cooking loss, L*, a*, b*, C*, H^0^, FA profiles of intramuscular fat, and volatile compounds in meat; µ is the least squares mean value; FR_i_ is the fixed effect of feeding regimes (i = 1: DOP-0; i = 2: DOP-40; i = 3: DOP-80); and e_ik_= random residual effect.

The goats’ proximal chemical composition, ME, and FA composition for milk during the suckling period were analysed with the repeated measures procedure. The model included the fixed between-subjects factor of dietary treatment, the fixed within-subjects factor of the week of lactation (repeated measures), and the interactions between these factors. The linear model used for each parameter was as follows:Yijk = μ + Ti + Aij + Wk + (T × W)ik + εijk
where Yijk = observations for dependent variables; μ = overall mean; Ti = effect of diet treatment; Aij = random effect of animal j for the i treatment; Wk = effect of the k week of lactation; T × W = interactions among these factors for the i treatment and k week of lactation, and εijk = random effect of residual. However and for simplification, the results for the factor week of lactation have not been presented in this paper.

For each sensorial property, we have computed the average value of the three evaluations provided by the three members of a family. This way, a data matrix is formed with 10 rows (10 families) and three columns (three feeding regimes). Therefore, this data matrix reflects a repeated measures design, as each family offers data to the three regimes. This design suggests the performance of a profile analysis [26] to evaluate the equality of the three population means for the three feeding regimes. The null hypothesis for this two-dimensional variable is that the three population means equal zero, which would mean that the three population means for the three feeding regimes are equal. This null hypothesis about the transformed variables has been tested with the HotellingsT2 function in the ICSNP R library. A random vector (X_1_, X_2_, X_3_) is defined where Xi denotes the average value of the three evaluations provided by the three members of a family for feeding regime i. Therefore, if we call µ1, µ2 and µ3 the population mean of Xi, the profile analysis will test the null hypothesis µ1 = µ2 = µ3. Previously, multivariate normality was assessed with the Mardia test, a generalisation of the univariate tests based on the skewness and kurtosis measures [26]. Both measures showed no statistical difference from the values assumed in the multivariate normality scenario. To identify different mean values, three comparisons of two matched samples were performed with the t-test function in R and a Bonferroni correction was considered (*p* < 0.05).

Finally, a linear discriminant analysis was conducted with variables that were significantly different to discriminate between the three feeding regimes. The linear discriminant analysis function in the MASS package in R was used [27].

## 3. Results and Discussion

There were no significant differences between the diet treatments in terms of birth weight, slaughter live weight, and cold carcass weight of suckling kids (on average 2.84 kg, 3.06 kg, and 2.95 kg; 8.98 kg, 9.03 kg, and 8.65 kg; 4.74 kg, 4.82 kg, and 4.65 kg, for DOP-0, DOP-40 and DOP-80, respectively; unpublished data). The cold carcass weights ranged from 4.65 kg to 4.82 kg and are in line with commercial carcass weights in the southern European market [16].

The mean values of chemical composition and FA profile of mother’s milk from two samples are shown in Table 2.

Mother diet had no significant effect on milk yield (the average daily milk yield and the total yield during the suckling period were 1.6 l/d and 60 l, respectively; unpublished data), chemical components, or ME, except for fat (*p* < 0.05; Table 2), which was significantly higher for the DOP-40 diet than for the DOP-80 diet (the DOP-0 diet was not significantly different than the other two diets). Although increased milk fat content is common when dietary fiber concentrations rise at the expense of starch [28], small differences were found in our study, probably due to the small differences in fibre content between the three diets. Furthermore, we observed significant effects of diet treatment on the FA profile (Table 2) for only two acids: C18:0 (*p* < 0.001; higher for the DOP-0 diet than DOP-80 diet, while the DOP-40 diet did not differ from the other two diets) and C20: 4 n-6 (*p* < 0.05; higher for DOP-40 diet than DOP-0 diet, while the DOP-80 diet did not differ from the other two diets). Concerning C18:0, the higher percentage of this acid in the DOP-0 diet (Table 1) may explain its increased abundance in DOP-0 milk.

### 3.1. Chemical Analysis and Physical Properties of the Meat

The chemical and physical characteristics of the meat are shown in Table 3. The pH values found were in line with several studies [14,29] and higher than those reported by Ripoll et al. [16] (pH values around 5.70) for European goat breeds. High ultimate pH values for goat muscles are common in the literature, suggesting that, unlike other species such as sheep and cattle, kids are generally highly prone to stress [30]. In the present study, no significant differences (*p* > 0.05) between the three treatments were observed in meat pH values after 24 hours. The use of DOP in the goat feed does not seem to affect the pH values of the kid meat, and so alterations to the properties of the meat during maturation are not expected.

The values for the chemical composition of the meat were within the ranges described by Guzmán et al. [14] in Payoya kids and Ripoll et al. [16] for kids of native Mediterranean goat breeds. The chemical analysis of the kid goat meat (Table 3) did not reveal significant differences between the treatments in terms of moisture, protein, fat, or ash content (*p* > 0.05). There were also no significant differences in the main descriptors of goat milk composition between the three diet groups (Table 2).

Meat tenderness is one of the most important attributes for consumer satisfaction. All the meat samples had shear force values in a similar range to the values reported by Marichal et al. [2] (5.5 to 8.1 kg/cm^2^) for suckling kid goats slaughtered at similar weights and fed exclusively with milk, which are representative of the kid goat market in Mediterranean Europe. The mother’s diet did not significantly affect shear force values in the kids’ LT muscles (Table 3). We found no significant difference (*p* > 0.05) between the groups in WHC, measured as expelled juice and cooking loss (Table 3). These observations are in line with Caparra et al. [7], who did not observe a dramatic effect on the main physical properties of meat after incorporating dried citrus pulp into the diet of lambs.

With regard to meat colour (Table 3), the meat was observed to have a reduced myoglobin content as compared to adult animals, as corresponds to suckling animals [29]. As the animals’ diet consisted exclusively of breast milk and this milk does not contain the iron necessary for myoglobin, it was expected that the meat would exhibit a reduced myoglobin level. In the present study, no differences between the treatment diets (*p* > 0.05) were observed in myoglobin content. Although orange pulp includes trace amounts of iron (0.009 mg/100 g), the lack of a dietary effect on myoglobin may reflect the absence of iron in mother’s milk, which makes it necessary to introduce other more iron-rich feeds (such as grass or cereal) to maintain meat myoglobin levels. The instrumental meat colour values were within the ranges observed by other authors using dairy goat kids slaughtered with similar weights [29] (ranges 50.07–56.93, 9.08–11.50 and 36.83–43.99 for L*, C* and H^0^ respectively), while they were lower than those reported in meat from heavier kids from meat goat breeds [16] (ranges 52.95–54.76, 18.05–23.65, and 44.30–59.90 for L*, C*, and H^0^ respectively). Mother’s diet significantly affected L* (*p* < 0.01), a*, and H^0^ (*p* < 0.05) values. The inclusion of DOP in the goats’ diet decreased the L* and H^0^ parameters of suckling goat meat, which were darker than those observed in the case of traditional diets. In fact, an increase in a* was observed in suckling goat meat when DOP was added to the goat feed. In general, the addition of DOP in the mother’s diet can lead to a decrease in the proportions of light pink meat and increase the proportion of darker meat in suckling goat meat.

### 3.2. Fatty Acids and Nutritional Properties of the Meat

While a total of 37 FAs were identified in the LT muscle, only the main FAs are shown in Table 4. In general, the results for the three treatments were within the ranges reported in kids of the Payoya breed [31]. The SFA group was the most prevalent (around 43% of total FAs detected), while MUFAs and PUFAs comprised approximately 32% and 25% of total FAs detected, respectively. Similar results have been reported in Spanish breeds of suckling kids [18].

There was no significant difference in SFA content between the treatments (*p* > 0.05), including major saturated palmitic FA (C16:0) (Table 4). Mother’s milk is rich in saturated short and medium-chain FAs (mainly C16:0). The use of concentrates rich in orange pulp in the dam’s ration had no effect on the C16:0 content of the suckled milk nor on the composition of the kids’ meat. Differences in the MUFA content of the kids’ meat (*p* < 0.05) were observed among the treatments. In fact, higher C18:1 fatty acid content in meat from the DOP-0 and DOP-40 diets (*p* < 0.05; Table 4) were observed. However, we did not observe significant changes in the content of the main unsaturated FA (C18:1) in mother’s milk when orange pulp was included in the goats’ diet (Table 2). Our results are in line with those of Wang et al. [32], who found higher C18:1 content in the meat of animals raised on cereal-based diets. On the other hand, the lower C18:0 content observed in the meat from DOP-80 kids as compared to the DOP-0 group (Table 4) reflects the fact that the milk of the DOP-80 group had the lowest C18:0 content (Table 2). This observation relates to the idea that the fat composition of the meat of lactating kids depends mainly on the characteristics of the milk they ingest because the ruminal processes of fat biohydrogenation are not evident in lactating animals.

In all the proposed treatments, the PUFA content in the kids’ meat was lower than the SFA and MUFA content (Table 4). This observation was expected because the PUFA content of the milk ingested was lower (Table 2) than the SFA and MUFA content. The PUFA content is important in meat because these FAs are highly oxidable during cooking [33]. Moreover, previous studies have indicated that SFAs (e.g., C16:0 and C18:0) and MUFAs (e.g., C18:1) are positively correlated with meat flavour [34], while PUFAs are negatively correlated. The effect of including DOP in the dam diets was insignificant for most PUFAs in the kid goat meat. Only two PUFAs were affected by the diet treatments. In fact, significant decreases of the most important PUFAs, C18:2 (*p* < 0.01) and C20:3 n-6 FA (*p* < 0.05) were observed in meat from diets that included orange pulp.

Consumers are increasingly interested in the lipid content of edible meat due to its relationship to human health. In a healthy human diet, the PUFA/SFA ratio should be as low as possible to reduce the risk of heart disease, among other health problems [35]. In our study, significant differences were observed in the PUFA/SFA ratio (*p* < 0.01; Table 4). The most favourable PUFA/SFA ratio for human health (0.60 index) was observed in the case of kid goat meat from treatments using the highest concentration of DOP in the dam’s diet (DOP-80). The reduced C18:0 content observed in kid goat meat from the DOP-80 treatment (Table 4) improved the PUFA/SFA ratio. Replacing cereal with DOP reduces the C18:0 content of the meat and therefore reduces the PUFA/SFA ratio.

A relationship between human health and the n-6/n-3 PUFA index has been proposed. Some clinical studies have recommended a n-6/n-3 ratio of less than four in order to reduce the risk of coronary disease [35]. The use of DOP in goat feed had a significant positive effect on the n-6/n-3 ratio (Table 4). The values obtained for the n-6/n-3 ratio in the three diet treatments (range 7.0 to 8.4) were higher than those recommended to prevent coronary heart disease. However, replacing cereal with DOP improved the n-6/n-3 ratio (*p* < 0.01) of kid goat meat, which may be due to the decrease of the most prevalent n-6 PUFAs, C18:2 and C20:3 n-6 FA, observed in meat from diets using orange pulp. Recommendations to consume foods with reduced thrombogenicity indices have also been suggested as a measure to improve human health [19]. The thrombogenicity index in the meat of the kids from DOP treatments was lower than that observed in the meat of kids whose goats ingested mainly cereal concentrate (DOP-0) (Table 4). This observation is related to the lower content of C18:0 and the higher content of C20:4 n-6 in the milk of the DOP-40 and DOP-80 mothers (Table 2).

### 3.3. Volatile Compounds and Aromatic Properties of Meat

The flavour of cooked meat play a major role in determining consumers’ acceptance of meat and their meat preferences [36]. As is reported in the literature, the analysis of volatile compounds in raw meat is complicated by the low percentage of intramuscular fat distributed heterogeneously in meat [37], as volatile compounds are stored in muscle fat only at trace levels. This observation is especially important for suckling kid goat muscle, as the average fat content is usually less than 2% in fresh meat. A total of 21 volatile compounds were tentatively identified in grilled goat kid meat for the three diet treatments (Table 5).

All the volatiles that we identified were clustered in the following chemical families: aldehydes (3), aliphatic ketones (4), aliphatic alcohols (1), furans (1), sulphur compounds (4), aliphatic aldehydes (5), aliphatic hydrocarbons (2), and aromatic hydrocarbons (1). Aldehydes were the main chemical family in cooked kid loins (around 55%), followed in decreasing order by aliphatic ketones, sulphur compounds, aliphatic aldehydes, aliphatic alcohols, furans, aromatic hydrocarbons and aliphatic hydrocarbons (around 19%, 13%, 11%, 0.8%, 0.7%, 0.4%, and 0.1%, respectively). The main chemical families of the volatile compounds found in this study are in accordance with those reported for cooked meat in the literature [40]. The generation of these volatile compounds in cooked meat is dependent upon the concentration of carbohydrates, amino acids, and lipids in the raw meat and the cooking conditions [37].

The effect of including orange by-products in goat feed was significant in a low number of individual volatile compounds (Table 5), including ethyl furan, dimethyl disulphide, and heptane. No significant differences were observed in other carbonyl compounds, such as unsaturated aldehydes or aliphatic ketones, among the three treatment groups. This observation reflects the small contribution of lipid oxidation to the development of aldehydes and ketones in suckling kid goat meat because of the meat’s low fat content.

Including orange pulp in the dams’ feed caused significant differences (*p* < 0.05) in the ethyl furan content of kid meat (Table 5), as the highest values were observed in kid steaks from DOP-40 and DOP-80 (0.20 and 0.30 AU × 10^6^). Furan is mainly associated with an aroma described as sweet and caramel-like [36]. Limacher et al. [41] found that the main formation pathways for furans are from the intact sugar skeleton of fruit. Under the roasting conditions used in our study, furans would mainly be formed from the intact sugar skeleton (mainly hexoses) of the orange pulp. Including DOP in goat feed also affected the meat’s sulphur compounds, causing significant differences (*p* < 0.05) in dimethyl disulphide in cooked suckling kid goat meat (Table 5), with the highest amounts of dimethyl disulphide occurring in kid goat steaks from DOP-80 (2.32 AU × 10^6^). While DOP is rich in sulphur compounds (0.11%/dry weight), the sulphur compound content of cereals varies [42]. Sulphur compounds were found to contribute to the meat odour of lambs [43], but there are no cases in the literature regarding sulphur compounds in suckling kid goat meat. Farmer et al. [44] found that sulphur compounds in meat are products of reactions involving sulphur-containing amino acids. Therefore, it is to be expected that the meat of kids from goats fed with DOP would have a higher content of sulphur-containing amino acids than the meat of kids from goats fed using cereal diets.

Madruga et al. [45] reported that hexanal is an important aldehyde in the aroma of goat meat. Aldehydes are derived from the lipid-oxidation Maillard reaction and the breakdown of amino acids through transamination and decarboxylation [46]. In fact, during cooking, hexanal is derived from the thermal degradation of C18:1. The lower C18:1 content observed in the meat of the goats from the DOP diet groups does not seem to have resulted in differences in meat hexanal content between the three treatment groups. A significant decrease in C18:1 content (*p* = 0.015; Table 4) was observed when increasing DOP in the ration of the mothers, while diet treatment was not observed to have a significant effect on hexanal content (*p* > 0.05; Table 5). This observation may relate to the low fat content in the meat of the suckling kids in all three treatment groups.

Aliphatic hydrocarbons are especially important for the typical aroma of meats and contribute to the characteristic flavour of several animal species [47]. With regard to these compounds, the highest heptane content (*p* < 0.05, Table 5) was observed in suckling kid goat meat from DOP-80. Elmore et al. [48] suggested that a diet high in PUFAs leads to an increase in heptane in grilled lamb meat. However, this relationship was not evident in the case of the kid goats used in the present study, since no significant effect of PUFA content in mother’s milk on heptane levels was observed (Table 2). The three diet treatments also had no significant effect on PUFA content in kid meat. It is likely that these results are related to the low fat content of the kid meat.

### 3.4. Sensorial Properties of Meat

The untrained panellists’ perceptions of the acceptability of suckling kid goat meat depending on the mother goat’s diet are shown in Table 6. In general, the scores ranged between seven and eight on a 10-point scale measuring tenderness, juiciness, flavour quality, and overall appraisal of the meat.

There was no specific preference between diet treatments concerning the tenderness or juiciness of the meat because the animals were young and the meat in all the treatments was tender and juicy (*p* > 0.05, Table 6). This result may be related to the fact that no significant differences were observed among the three treatments for the shear force or water losses of the meat (Table 3). However, the flavour quality of the meat was influenced by the dam’s feed (*p* < 0.01). Meat from treatments including DOP was considered to have a more desirable flavour than the meat from the cereal-based diet (Table 6). There was no general agreement regarding the desirability of the flavours. There was considerable disagreement among the subjects about this factor. However, consumers from a specific geographic area tended to like the same kind of meat, probably due to the influence of previous experiences or the gastronomic culture in local areas, which are partially determined by the products available on the market [24].

Fat content can affect the aromatic characteristics of meat. However, no significant differences were observed among the three diet treatments in terms of the fat content of suckling kid goats (Table 3). Accordingly, the fat content of the meat should not be considered to have been a factor in the untrained panellists’ evaluation. Alternatively, the higher flavour values likely relates to the specific content of aromatic compounds detected in the meat. Resconi et al. [49] identified several volatile sulphur compounds that contribute to the development of meat flavour. In our study, the highest content of dimethyl disulphide was observed in kid goat meat from treatments that included DOP in the goat feed (Table 5). Dimethyl disulphide is a volatile compound derived from the oxidation and degradation of cysteine and methionine (sulphuric amino acids) that can be aromatically detected at low concentrations [49]. Ethyl furan and aliphatic hydrocarbons such as heptane could also contribute to the differences in flavour quality reported by the untrained panellists (these compounds occurred at the highest levels in meat from the DOP treatments).

Many individual sensory attributes influence the acceptability of meat (e.g., juiciness, colour, greasiness, etc.). However, an overall appraisal provides an idea of whether the consumers liked the meat. The effect of including DOP in the dam’s feed on the overall acceptability of kid goat meat was insignificant (Table 6). The low fat content in suckling kid goats is likely to have influenced this assessment for untrained panellists.

### 3.5. Discriminant Analysis

A linear discriminant analysis model was built to determine the relationship between groups of variables and the three feeding groups for datasets that were significantly different of suckling kid goat meat (Figure 1).

We performed a discriminant analysis to quantify the contribution of each variable to the differences observed among the three treatments. Since the number of variables analysed is very large, only those variables that were significantly different (*p* < 0.05) according to a permutation ANOVA test were selected in the model. The resulting model was used to identify the key variables that contribute to the differences in the meat suckling kid goats based on meat quality parameters, FA profile, and volatile compound content. Function 1 of the discriminant analysis accounted for 72.9% of the total variation among feeding strategies and it was mainly determined by the absence (DOP-0) or presence of DOP in goat feed (DOP-40 and DOP-80). This function clearly discriminated the effect of including DOP in goat feed, with meat from goats fed using DOP on the right and meat from goats fed with concentrate without DOP on the left. The parameters of colour L*, a*, and hue angle determined the variability assigned to LD1 (Table 3).

On the other hand, Function 2 discriminated three groups (27.0%) taking into account the level of DOP included in the feed. The high-DOP concentrate group (DOP-80) was in the upper quadrant with the highest aromatic compound content (i.e., ethyl furan, dimethyl disulphide and heptane; Table 5), whereas the groups of concentrates not including DOP (DOP-0) and with low DOP (DOP-40) were in the lower quadrant with the highest monounsaturated FA content (Table 4). The ability of LDF1 and LDF2 to separate the three groups suggests that using DOP in the feeding regime of goats could affect the kids’ meat colour, the prevalence of some volatile compounds (ethyl furan, dimethyl disulphide, and heptane), and MUFA content.

## 4. Conclusions

We can recommend replacing from 40% to 80% of cereals with dehydrated orange pulp (DOP) in the diet of mother goats, given that it is unlikely that this will cause significant changes in the physical and chemical properties of the kids’ meat. Moreover, including DOP in dams’ diets improves untrained panellists’ appraisal of the sensory characteristics of the kids’ meat. This is mainly due to flavour quality, which can be attributed to the increased development of volatile compounds such as ethyl furan, dimethyl disulphide, and heptane observed in kid goat meat obtained from goats fed with DOP. In addition, including DOP in goat diets may improve the nutritional indices of fatty acids in kid goat meat for human health, since it contributes to the reduction of n-6/n-3 and thrombogenicity indices and an increase of the PUFA/SFA index. C18:1 FA and certain volatile compounds (furans, sulphur compounds, and aliphatic hydrocarbons) may be considered to be the main discriminators for kid goat meat obtained by including DOP in goat feed. From the perspective of sustainable development, DOP is available as an orange industry by-product and represents a plausible strategy for improving the profitability of goat production for farmers.

## Figures and Tables

**Figure 1 animals-10-00766-f001:**
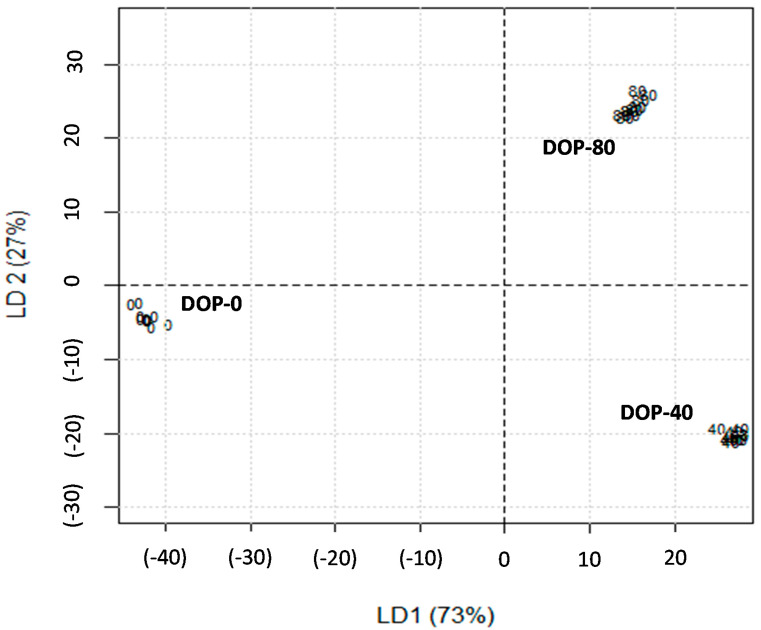
Analysis of observed variables that discriminate the meat of suckling kid goats from three dam diet treatments: commercial concentrates with alfalfa hay (DOP-0), concentrate including 40% of replacement of cereals by pellets of dehydrated orange pulp and alfalfa hay (DOP-40), and concentrate including 80% of replacement of cereals by pellets of dehydrated orange pulp and alfalfa hay (DOP-80).

**Table 1 animals-10-00766-t001:** Ingredients, chemical composition, nutritive value, and proximate fatty acid (FA) composition of the experimental diets ^1^ used to feed the Payoya goats.

Ingredients (% Dry Matter Basis)	DOP-0	DOP-40	DOP-80
Alfalfa hay	17.44	17.53	17.64
Concentrate			
Dehydrated orange pulp (pellets)	0.00	19.96	39.87
Grain (oats)	22.10	13.23	4.38
Grain (barley)	8.53	5.11	1.70
Grain (corn)	19.34	11.61	3.89
Soy flour 44%	7.31	10.23	12.97
Sunflower pellets 28%	12.84	12.50	13.78
Grain (peas)	10.32	8.12	4.05
Salt	0.41	0.41	0.41
Stabilised butter	0.41	0.00	0.00
Vitamins and minerals ^2^	1.31	1.31	1.32
**Chemical Composition (% Dry Matter Basis)**			
Dry matter (DM)	90.51	89.82	89.97
Crude protein	17.76	16.59	18.43
Crude fibre	9.27	10.62	12.33
Ether extract	3.42	2.31	1.71
Ash	4.82	6.44	7.50
Forage unit for lactation (UFL/kg DM)	0.98	0.98	0.97
Protein digestible in the intestine (PDI)	10.5	11.0	11.5
**Proximate Fatty Acid Composition (%)**			
C8:0–C14	1.54	3.12	4.54
C16:0	31.4	23.6	26.3
C16:1	1.04	0.63	0.62
C18:0	11.8	9.70	10.7
C18:1 n-9 cis	34.1	23.3	19.3
C18:2 n-6 cis	48.3	34.8	28.2
C18:3 n-6	0.18	0.18	0.44
C18:3 n-3	2.92	3.81	4.12
∑SFA	44.7	36.4	41.5
∑MUFA	35.1	24.0	19.9
∑PUFA	51.4	38.8	32.8

^1^ DOP-0, a diet composed of commercial concentrates and alfalfa hay; DOP-40, a diet composed of concentrate and alfalfa hay with 40% of concentrate cereals replaced by pellets of dehydrated orange pulp; and DOP-80, a diet composed of concentrate and alfalfa hay with 80% of concentrate cereals replaced by pellets of dehydrated orange pulp. ^2^ Nutral cabras LD granulado, Cargill^®^. SFA: saturated fatty acids; MUFA: monounsaturated fatty acids; PUFA: polyunsaturated fatty acids; UFL: Forage Unit for Lactation.

**Table 2 animals-10-00766-t002:** Chemical composition and fatty acid (FA) content (expressed as mg/g dry matter) of mother’s milk (mean values of the two samples) during the natural suckling period by experimental diet type ^1^.

Item ^3^	DOP-0	DOP-40	DOP-80	SEM ^2^	*p*-Values
No. of goats	10	10	10		
Dry matter (%)	13.14	12.95	12.56	0.254	0.757
Fat matter (%)	4.94 ^a,b^	5.33 ^a^	4.31 ^b^	0.135	0.027
Crude protein (%)	3.89	4.20	3.82	0.094	0.107
ME (MJ/kg)	3.44	3.54	3.25	0.054	0.108
∑ SFA	206.39	207.17	201.79	2.611	0.401
C8:0-C13:0	56.35	58.92	58.49	0.836	0.370
C14:0	29.80	30.53	29.58	0.453	0.252
C16:0	69.90	70.20	68.67	0.928	0.672
C18:0	31.49 ^a^	29.68 ^a^^,^^b^	27.28 ^b^	0.505	0.001
∑ MUFA	70.45	68.85	64.89	1.026	0.083
C16:1	3.36	3.47	3.36	0.087	0.657
Total C18:1	64.64	63.07	59.33	0.972	0.177
∑ PUFA	16.83	16.29	15.60	0.311	0.314
Total C18:2 n-6	11.45	11.15	10.86	0.280	0.736
C18:3 n-3	0.91	0.94	0.94	0.027	0.580
C20:2	0.15	0.16	0.15	0.005	0.160
C20:3 n-6	0.18	0.18	0.16	0.007	0.079
C20:4 n-6	0.08 ^b^	0.10 ^a^	0.09 ^a,b^	0.003	0.014
C20:5n-3	0.36	0.35	0.35	0.006	0.784
C22:5n-3	0.40	0.40	0.37	0.009	0.264
C22:6n-3	0.41	0.39	0.36	0.011	0.323
Others FA	24.20	22.78	22.30	0.417	0.204
PUFA/SFA	0.08	0.08	0.08	0.001	0.343
n-6/n-3	5.42	5.27	5.29	0.166	0.552
AI	2.34 ^b^	2.47 ^a,b^	2.55 ^a^	0.023	0.101
TI	2.72	2.76	2.80	0.020	0.390

Different superscripts (^a,b^) indicate significant differences (*p* < 0.05) among treatments. ^1^ See Table 1; ^2^ Standard error of mean; ^3^ SFA: saturated fatty acids; PUFA: polyunsaturated fatty acids; AI: atherogenicity index; TI: thrombogenicity index; ME: metabolisable energy.

**Table 3 animals-10-00766-t003:** Chemical composition of *triceps brachii* muscle and instrumental traits of *longissimus* muscle of suckling goat kids, according to the use of dehydrated orange pulp in feeding dams ^1^.

Chemical and Physical Characteristics	DOP-0 (n = 10)	DOP-40 (n = 10)	DOP-80 (n = 10)	SEM ^2^	*p*
pH_24hours_	5.93	5.89	6.10	0.045	0.219
Chemical composition (% fresh meat)					
Moisture	77.13	77.48	77.12	0.239	0.843
Protein	18.68	18.54	18.57	0.195	0.981
Fat	1.99	1.83	1.95	0.059	0.486
Ash	1.24	1.28	1.34	0.034	0.545
Shear force (kg/cm^2^)	5.93	5.36	6.56	0.213	0.074
Water loss (% fresh meat)					
Expelled juice	17.13	16.47	16.15	0.407	0.630
Cooking loss	25.76	25.68	24.04	0.648	0.492
Colour					
Myoglobin content (mg/g fresh meat)	2.42	2.23	2.63	0.169	0.520
Lightness (L*)	56.19 ^a^	53.94 ^b^	53.14 ^b^	0.424	0.004
Redness (a*)	4.52 ^b^	6.30 ^a^	5.83 ^a^	0.237	0.029
Yellowness (b*)	7.40	6.06	6.66	0.282	0.054
Chroma (C*)	8.80	9.23	8.61	0.256	0.367
Hue angle (^0^)	58.03 ^a^	44.81 ^b^	48.26 ^b^	1.856	0.013

Different superscripts (^a,b^) indicate significant differences (*p* < 0.05) among treatments. ^1^ See Table 1; ^2^ SEM: standard error of the mean.

**Table 4 animals-10-00766-t004:** Fatty acid (FA) composition of the meat of suckling goat kids (expressed as mg FA/100g fresh meat) according to the use of dehydrated orange pulp in feeding dams ^1^.

Item ^2^	DOP-0 (n = 10)	DOP-40 (n = 10)	DOP-80 (n = 10)	SEM ^3^	*p*
∑ SFA	602.36	579.87	523.00	14.300	0.060
C8:0–C13:0	7.70	7.87	6.69	7.415	0.389
C14:0	40.33	40.51	35.70	1.864	0.506
C16:0	298.92	290.67	269.73	8.146	0.332
C18:0	229.56 ^a^	213.79 ^a^	186.41 ^b^	4.602	0.000
∑ MUFA	483.3 ^a^	479.8 ^a^	456.5 ^b^	13.983	0.018
C16:1	21.40	23.37	18.67	0.985	0.148
Total C18:1	404.6 ^a^	393.3 ^ab^	371.6 ^b^	12.648	0.015
∑ PUFA	344.63	318.23	316.95	6.405	0.137
Total C18:2 n-6	159.27 ^a^	135.69 ^b^	139.73 ^b^	3.526	0.009
C18:3 n-3	4.72	5.10	4.50	0.127	0.148
C20:2	21.12	19.79	18.03	0.668	0.167
C20:3 n-6	6.63 ^a^	5.81 ^b^	5.68 ^b^	0.155	0.020
C20:4 n-6	108.55	106.64	107.36	2.637	0.959
C20:5 n-3	5.20	5.67	4.97	0.205	0.059
C22:5 n-3	16.31	16.18	16.52	0.283	0.891
C22:6 n-3	6.65	7.48	6.92	0.237	0.361
Others FA	214.90 ^a^	191.96 ^ab^	189.32 ^b^	4.519	0.033
Ratios to nutritional human health					
PUFA/SFA	0.57 ^b^	0.55 ^b^	0.61 ^a^	0.786	0.007
n-6/n-3	8.39 ^a^	7.06 ^b^	7.58 ^b^	0.184	0.007
AI	0.59	0.59	0.61	0.009	0.660
TI	1.26 ^a^	1.19 ^b^	1.20 ^b^	0.011	0.010

Different superscripts (^a,b^) indicate significant differences (*p* < 0.05) among treatments. ^1^ See Table 1; ^2^ SFA: saturated fatty acids; MUFA: monounsaturated fatty acids; PUFA: polyunsaturated fatty acids; AI: atherogenicity index; TI: thrombogenicity index. ^3^ SEM: standard error of the mean.

**Table 5 animals-10-00766-t005:** Volatile compounds of the meat of suckling goat kids (expressed in area unit × 10^6^) according to the use of dehydrated orange pulp in feeding dams ^1^.

Volatile Compounds	I_DB-5_	MS	RI	DOP-0 (n = 10)	DOP-40 (n = 10)	DOP-80 (n = 10)	SEM	*p*
**Aldehydes**								
Acethaldehyde	<500	+	+	24.37	28.44	30.77	1.404	0.230
2-Methyl butanal	648	+	+	5.62	6.31	7.40	0.500	0.678
3-Methyl butanal	657	+	+	9.84	10.22	11.57	0.801	0.652
Total (%)				54.44	55.80	54.97		
**Aliphatic Ketones**								
2-Propanone	503	+	+	11.15	11.23	13.13	0.565	0.512
2,3-Butanedione	593	+	+	0.18	0.17	0.19	0.019	0.864
2-Butanone	601	+	+	2.83	3.17	3.96	0.274	0.500
n-Nonane	622	+		0.33	0.33	0.49	0.041	0.266
Total (%)				19.80	18.49	19.64		
**Aliphatic Alcohols**								
Ethanol	669	+	+	0.61	0.67	0.80	0.100	0.857
Total (%)				0.83	0.83	0.88		
**Furans**								
Ethyl furan	955	+	+	0.14 ^b^	0.20 ^a,b^	0.30 ^a^	0.027	0.036
Total (%)				0.19	0.25	0.33		
**Sulphur Compounds**								
Methanethiol	1280	+	+	7.12	8.28	8.30	0.548	0.953
Dimethyl disulfide	761	+	+	1.20 ^c^	1.46 ^b^	2.32 ^a^	0.194	0.047
Carbon disulfide	544	+	+	0.18	0.20	0.18	0.009	0.142
2,3,4-Trisulfide	970	+	+	0.81	0.80	0.76	0.130	0.961
Total (%)				13.09	13.05	12.83		
**Aliphatic Aldehydes**								
2-Methyl propanal	905	+	+	7.46	8.28	8.96	0.550	0.895
Hexanal	800	+	+	0.30	0.35	0.45	0.050	0.415
Heptanal	900	+	+	0.22	0.14	0.14	0.027	0.665
Octanal	1002	+	+	0.13	0.14	0.14	0.027	0.815
Nonanal	1109	+	+	0.05	0.07	0.04	0.018	0.764
Total (%)				11.15	11.14	10.75		
**Aliphatic Hydrocarbons**								
Heptane	700	+	+	0.06 ^b^	0.07 ^b^	0.16 ^a^	0.014	0.043
3-Methyl heptane	783	+		nd	nd	0.02	0.023	-
Total (%)				0.08	0.07	0.18		
*Aromatic Hydrocarbons*								
Methyl benzene	667	+	+	0.29	0.29	0.37	0.028	0.881
Total (%)				0.40	0.36	0.41		

Different superscripts (^a,b^) indicate significant differences (*p* < 0.05) among treatments; nd: not detected. *I*_DB-5_: retention index as determined using hydrocarbons on an HP-5 fused silica column. MS: tentatively identified by matching the simple spectrum against the Wiley 275 library spectrum; RI: approximate identification by comparing the retention index with literature values [38,39]; ^1^ See Table 1. SEM: standard error of the mean.

**Table 6 animals-10-00766-t006:** Evaluations of the tenderness, juiciness, flavour quality, and overall quality of the meat of suckling goat kids by untrained panellists, according to the use of dehydrated orange pulp in feeding dams ^1^.

Sensory Attributes	DOP-0 (n = 10)	DOP-40 (n = 10)	DOP-80 (n = 10)	SEM	*p*
Tenderness	7.37	7.43	7.56	0.548	0.344
Juiciness	7.19	7.40	7.53	0.414	0.171
Flavour quality	7.10 ^b^	7.53 ^a^	7.77 ^a^	0.606	0.007
Overall appraisal	7.17	7.59	7.72	0.619	0.074

Different superscripts (^a,b^) indicate significant differences (*p* < 0.05) among treatments. ^1^ See Table 1. SEM: standard error of the mean.

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
