# Peer review of "The Use of Concentrates Rich in Orange By-Products in Goat Feed and Its Effects on Physico-Chemical, Textural, Fatty Acids, Volatile Compounds and Sensory Characteristics of the Meat of Suckling Kids"

_animals, 2020, doi:10.3390/ani10050766_

Round 1
Reviewer 1 Report
Dear authors,
The idea of using orange by-product as DOP for animal production is wonderful and in line with the new trends in food sustainability. Furthermore, I understand your efforts by analyzing the meat of suckling kid by different methods including physico-chemical, textural, fatty acids, volatile compounds and sensory characteristics.
However, there is no sense in measure differences in meat from suckling kids when there is no difference in their feeding (milk composition, table 2), genetic and management system. I mean, the presence of DOP in feed will change the composition of milk or meat of the animal that is eating that feed (calving goats), but if there is no differences in milk (table 2), there is no sense to think that will be differences in the meat of breastfeeding kids.
Here I explain some other flaws:
- Materials & methods section is incomplete: for example, there is no information about the Payoya breed (meat or milk aptitude) and the usual reared system (intensive, extensive, etc.), which will vary the influence of the feed in the milk production and composition. On the other hand, the substitution of about 80% of concentrate by DOP will probably change the taste of the feed, but nothing related to that is mentioned. The information about milk sampling is not enough. This information is very important taking into account that milk composition vary significantly during the first days of suckling and that kids are exclusively breastfeeding.
- Materials & method related to FA composition: Milk and meat fatty acid (FA) analysis are realized under different methods and there is no mention of why. In milk FA composition, there is no mention of how many FA have been unidentified in milk, and what FA are considered in each of FA main groups (table 2). 37 FA in meat (line 271) is not representative of an analysis with 120m GC column (line 148) and is indicative of an incorrect quantification in table 4. As the FA presented in table 2 related to milk are not the same presented in table 4, that impede the correct analysis of the meat data.
- Table 2 data are not correct: fat matter and crude proteins data seem to be in g per 100g of milk, not per 100g of DM.
With your consideration, I will recommend the following options:
1) To study the influence of DOP in calving goats’ physiological components (if there is influence in the quantity of feed they eat -sensory properties-, if there is a gain or not of weight, etc), and their products (milk or meat) under proper analysis. For that, I would realize a complete analysis of three feeding systems, including FA analysis; and I would explain with all details the management systems and the breed characteristics.
2) To study the meat characteristics (all the have measure in this paper) of suckling kids. For that, the methods would be better explained and the data would be analysed with more precision. In this case, it will be positive not to found differences relative to the calving goats feeding system: the stockbreeder can found an economical profit using orange by-products, under an ecological point of view is also a benefit and from a sensory point of view the lack of differences could be positive because consumers are used to eat meat from intensive production system where concentrate percentage is higher than in the proposed DOP80.
3) To study the differences in volatile composition ok milk from calving goats related to the substitution of concentrate with DOP in their feeding system. This new analysis linked with volatile composition of meat from suckling kids and sensory analysis of consumers, could give an interesting paper.
Hope my revision is helpful for you.
Author Response
ANSWER TO THE REVIEWER 1
However, there is no sense in measure differences in meat from suckling kids when there is no difference in their feeding (milk composition, table 2), genetic and management system. I mean, the presence of DOP in feed will change the composition of milk or meat of the animal that is eating that feed (calving goats), but if there is no differences in milk (table 2), there is no sense to think that will be differences in the meat of breastfeeding kids.
We are in agreement with that comment. This is precisely one of the specific objective of the work. The authors aims to demonstrate the consequences of the use different concentrations of milk in the goat DOP feeding of kids meat quality. Thus, the focus is to check whether the inclusion of orange pulp in the ration of the mother has an impact on the quality of the kids meat. Based on the findings of the work, it is concluded that there are no great differences between the quality of the kids meat taking into account the feeding of the kids (only for one month) has not undergone major changes in chemical composition parameters of milk (Dry matter, fat matter and crude protein) and fatty acid profile (See table 2). That result may have important consequences for the farmer who belongs to DOP, since the farmer can use this by-product in goat diets without substantially altering the quality of the kids meat.
Here I explain some other flaws:
- Materials & methods section is incomplete: for example, there is no information about the Payoya breed (meat or milk aptitude) and the usual reared system (intensive, extensive, etc.), which will vary the influence of the feed in the milk production and composition. On the other hand, the substitution of about 80% of concentrate by DOP will probably change the taste of the feed, but nothing related to that is mentioned. The information about milk sampling is not enough. This information is very important taking into account that milk composition vary significantly during the first days of suckling and that kids are exclusively breastfeeding.
Reference to the Payoya breed has been included in the introduction section. See last paragraph of introduction. (Lines 75 and 76)
Changes in the composition of the rations had no influence on the palatability of the meat. After birth, feed data offered and refused were recorded daily for each group, and the intake per animal was calculated (data not show because they are under review in the 'animal' magazine). New text was included in 2.1. Animal management and dietary treatments. (Lines 94-95)
Information about milk sampling has been included in 2.1. Animal management and dietary treatments. (Lines 95-100)
- Materials & method related to FA composition: Milk and meat fatty acid (FA) analysis are realized under different methods and there is no mention of why.
The authors have been working on the fatty acid profile of meat and for a long time. They have found that the method proposed by Sukhija and Palmquist (1988) to identify fatty acids in milk is reliable. On the other hand, the authors have been carrying out the identification of fatty acids in meat based on the method proposed by Aldai et al. (2006) having good results. (Lines 172-186) Both methods was reported by Gutiérrez et al. (Line 173) and Horcada et al. (Line 178)
In order to a better interpretation of the results, tables 2 and 4 have been modified. Table 2 presents results in mg / g Dry matter and Table 4 presents results in mg FA / 100g fresh meat
In milk FA composition, there is no mention of how many FA have been unidentified in milk, and what FA are considered in each of FA main groups (table 2). 37 FA in meat (line 271) is not representative of an analysis with 120m GC column (line 148) and is indicative of an incorrect quantification in table 4. As the FA presented in table 2 related to milk are not the same presented in table 4, that impede the correct analysis of the meat data.
Thanks for your suggestion. Also, another reviewer suggests that Tables 2 and 4 include the same fatty acids under study. This has been done. Tables 2 and 4 have been constructed again indicating the same fatty acids for discussion.
Several works have been reported in the literature about the convenience of using 120 m GC column to identify 37 fatty acid in meat (supelco 37) (Horcada et al., 2012; Bravo-Lamas et al., 2018).
Horcada A., Ripoll G., Alcalde M.J., Sañudo C., Teixeira A. and Panea B. (2012). Fatty acid profile of three adipose depots in seven Spanish breeds of suckling kids. Meat Science, 92: 89-96.
Bravo-Lamas L., Barron L., Farmer L., Aldai N. (2018). Fatty acid composition of intramuscular fat and odor-active compounds of lamb commercialized in northern Spain. Meat Science, 139: 231-238.
- Table 2 data are not correct: fat matter and crude proteins data seem to be in g per 100g of milk, not per 100g of DM.
This is true. In table 2, units have been expressed in % of fresh milk
With your consideration, I will recommend the following options:
1) To study the influence of DOP in calving goats’ physiological components (if there is influence in the quantity of feed they eat -sensory properties-, if there is a gain or not of weight, etc), and their products (milk or meat) under proper analysis. For that, I would realize a complete analysis of three feeding systems, including FA analysis; and I would explain with all details the management systems and the breed characteristics.
The authors include reference to milk production in the three goat feeding groups at the beginning of section 3. (Results and discussion Lines 286-289). More information about these results is currently under review for publication in the journal Animal (currently under 'Major review'), so references to goat production in this document are not expanded. FA analysis milk between treatments has been reported because a new Table 2 has been included in the text.
2) To study the meat characteristics (all the have measure in this paper) of suckling kids. For that, the methods would be better explained and the data would be analysed with more precision. In this case, it will be positive not to found differences relative to the calving goats feeding system: the stockbreeder can found an economical profit using orange by-products, under an ecological point of view is also a benefit and from a sensory point of view the lack of differences could be positive because consumers are used to eat meat from intensive production system where concentrate percentage is higher than in the proposed DOP80.
In response to your suggestions, changes in the wording of the conclusions have been included (see '4. Conclusions' section)
3) To study the differences in volatile composition ok milk from calving goats related to the substitution of concentrate with DOP in their feeding system. This new analysis linked with volatile composition of meat from suckling kids and sensory analysis of consumers, could give an interesting paper.
Unfortunately, the authors are not able to analyze the volatile compounds in milk because they do not have a sample to carry out the studies. Only analyzes of the fatty acid profile of the milk of the mothers and the meat of the kids have been carried out because these compounds are considered precursors of the volatile compounds in meat. Volatile compounds are generated during the cooking of the meat (Vasta and Priolo, 2006). In the present work a description of the fatty acid composition of milk and meat is made (Tables 2 and 4).
Vasta V and Priolo A. (2006). Ruminant fat volatiles as affected by diet. A review. Meat Science, 73: 218-228

Reviewer 2 Report
line 46 what does tm mean?
table 1 review PDO percentages.
Author Response
ANSWER TO THE REVIEWER 2
line 46 what does tm mean?
We agree with ‘tm' was a typing error. 'tm' has been changed to 't' (unit of tonnes) (Line 52)
table 1 review PDO percentages.
We agree with the reviewer . There was a misprint in the table 1, the data was actually expressed in dry matter basis. For this reason, Table 1 has been corrected.

Reviewer 3 Report
Comments for Manuscript ID: animals-754966
The main concern is the novelty of the study which should be better explained as there are many studies on dried orange pulp. The presentation of both results and discussion were poorly done as there is lack of mechanism of action to support the results in the discussion. Furthermore, DOP has high content of prooxidant fatty acid, flavonoids, total phenols, Vitamin C and E, and carotenoids which could have been analysed both in feed ingredients, diets and in meat samples , and this could have strengthened the study especially when discussing the differences between treatments on fatty acid profiles, volatile compounds and sensory attributes. There are several inconsistencies in the results presented in tables.
Comment 1 Line 41: keywords should be written in alphabetical order.
Comment 2 Lines 59-60: add references
Comment 3 Line 67-71: Orange pulp has been studied extensively and in the present study there is no novelty.
Comment 4 Line 81 Fatty acids composition of orange pulp and diets needs to be presented. Diets are not balanced and there is need of ME to be added in Table 1 for both the DOP and all the experimental diets
Comment 5 Line 98: Preparation of fresh orange pulp, process and method used to dry the fresh orange pulp and its milling before diets formulation needs to be added before chemical analysis of diet. What did you substitute to formulate 40 DOP and 80 DOP?
Comment 6 Line 123-127: Did you perform the proximate composition in duplicates or triplicates?
Comment 7 Line 144-146: Briefly explain the procedure used for the determination of FA profile for both milk and meat
Comment 8 Line 161-163: There is need to mention the internal standard used before analysing volatile compounds on GC
Comment 9 Line 167-168: The formula used to compute the RI needs to be included
Comment 10 Line 184-188: Did you use the hedonic scale if so then you need to mention it
Comment 11 Line 189: Add the program and statistical software and models used to analyse the data
Comment 12 Line 238: Consider reporting WBSF and yellowness in your results as there is tendency
Comment 13 Line 246-249: Are your values falling within the recommended range? Please add the recommended range of goat meat tenderness and compare with other studies.
Comment 14 Line 255-257: Did you measure the iron content in the milk they ingested? Do not speculate describe what you did.
Comment 15 Line 259: You did not measure the Orange pulp iron. Furthermore, you incorporated the orange pulp in the diet you did not feed it direct to the animals so you should have measured the iron content of all your diets.
Comment 16 Line 264-269: Please add the range of values of the recommended lightness, redness yellowness, hue angle and chroma etc
Comment 17 Line 270: This section was poorly done, there is need to explain the mechanism of action of the active ingredients in DOP on fatty acid profiles. DOP has polyphenolic compounds like phenols, vitamin E carotenoids and vitamin C which have antioxidant and antimicrobial properties. The fact that you did not measure them in DOP and your diets makes it discussion of results difficult. You should also have analysed the quantity of vitamin E and total phenols in meat as well this could have made your results easy to discuss.
Comment 18 Line 322-327: Consider deleting these statements.
Comment 19 Line 371-372: Mechanism of action is lacking as well to support your results for example aldehydes are derived from lipid oxidation mallard reaction strecker degradation and breakdown of amino acid through transamination followed by decarboxylation. Hexanal are derived from oxidation of oleic acid whilst benzaldehyde is derived from linoleic acid and also from degradation of phenylalanine.
Comment 20 Line 379-381: Where is Table 6 in this manuscript. Explain the effects of DOP on juiciness and tenderness. There is need to explain the mechanism of action behind this.
Comment 21 Line 388-390: why?
Comment 22 Line 393-395: Dimethyl disulphide is a volatile sulphur compound which has a significant effect on meat flavour and aroma. This volatile compound is derived from oxidation and degradation of cysteine and methionine sulphur amino acids
Comment 23 Line 402-404: There is a need to explain the mechanism of action
Comment 24 Line 433: Overall, the conclusion, needs to be reworked
Comment 25 Line 437-439: Rewrite the conclusion.
Author Response
ANSWER TO THE REVIEWER 3
The main concern is the novelty of the study which should be better explained as there are many studies on dried orange pulp. The presentation of both results and discussion were poorly done as there is lack of mechanism of action to support the results in the discussion. Furthermore, DOP has high content of prooxidant fatty acid, flavonoids, total phenols, Vitamin C and E, and carotenoids which could have been analysed both in feed ingredients, diets and in meat samples , and this could have strengthened the study especially when discussing the differences between treatments on fatty acid profiles, volatile compounds and sensory attributes. There are several inconsistencies in the results presented in tables.
Authors thank the reviewer´ comments. The authors agree with the reviewer. This suggestion was considered by the authors before writing the document. In fact, they have antioxidant compounds analysis in the milk of the three treatments. These results are currently being evaluated in a major review in the journal 'Animal'. Authors consider not duplicating results in journals, therefore results of antioxidant agents in mothers' milk are not presented in this document.
However, the authors have previously made a reflection. With the objective of this work, we have tried to describe the quality characteristics of the goat meat using different levels of dehydrated orange pulp in three feeding systems. The authors have considered that in the present work the influence of antioxidant agents could be irrelevant taking into account the low slaughter weight of the kids. Besides, these animals have not ingested directly the feed that could include the antioxidant products. For this reason, unfortunately, no analysis of antioxidant agents have been carried out in the kid’s meat. In fact, kids have ingested only mother's milk for only one month of life.
Comment 1 Line 41: keywords should be written in alphabetical order.
Changes have been done (line 47)
Comment 2 Lines 59-60: add references
Reference Bampidis et al. (2006) has been included [4]. This is a review about Citrus by-products as ruminant feeds. (Line 65)
Comment 3 Line 67-71: Orange pulp has been studied extensively and in the present study there is no novelty.
We agree with this comment. There are works in the literature that analyze the use of citrus by-products in the production of pigs, and even of heavy lambs. However references in the breeding of kids have not been found. The authors consider the novelty of the present manuscript is increasingly important in the modernization of the goat meat production systems in the countries where this product is traditionally consumed.
Comment 4 Line 81 Fatty acids composition of orange pulp and diets needs to be presented. Diets are not balanced and there is need of ME to be added in Table 1 for both the DOP and all the experimental diets
Changes have been made in Table 1. The three mothers' diets were isoenergetic (Forage Unit for Lactation (UFL/kg DM) and isoproteic (Protein Digestible in the Intestine) were included in table 1 and the text (line 86). It is shown in the table 1. The profile of important fatty acids in goat rations is also shown.
Comment 5 Line 98: Preparation of fresh orange pulp, process and method used to dry the fresh orange pulp and its milling before diets formulation needs to be added before chemical analysis of diet. What did you substitute to formulate 40 DOP and 80 DOP?
The cereals in DOP-0 were replaced by the orange pulp (at 40 and 80% respectively). This observation was reported in ‘2.1. Animal management and dietary treatments’ section. Also, this substitution can be clearly seen in Table 1 of diets.
Comment 6 Line 123-127: Did you perform the proximate composition in duplicates or triplicates?
Analysis were performed in duplicate. Animals are small. The portion of muscle available is scarce. Not much sample is available for laboratory testing. So, measurements of proximal muscle composition are fairly stable overall. Reference was included in the text (Line 152)
Comment 7 Line 144-146: Briefly explain the procedure used for the determination of FA profile for both milk and meat
A brief explain the procedure used for the determination of FA profile for both milk and meat was included in '2.4. Fatty Acid Analysis' section (Lines 173-186).
Comment 8 Line 161-163: There is need to mention the internal standard used before analysing volatile compounds on GC
Internal standard in this experience was not used (Vasta et al., 2012). However, n-alkanes (C5 to C18) were included by previous injection to calculate RI index
Comment 9 Line 167-168: The formula used to compute the RI needs to be included
Description of the RI formula has been included in the text (section 2.5. Volatile Compound Analysis) (Lines 207-212)
Comment 10 Line 184-188: Did you use the hedonic scale if so then you need to mention it
The authors consider that the hedonic scale should be presented following the recommendations of other publications on food sensory content. (Lines 230-233)
Comment 11 Line 189: Add the program and statistical software and models used to analyse the data
Presentation of the statistical models has been clarified in 2.7. Statistical Analysis section
Comment 12 Line 238: Consider reporting WBSF and yellowness in your results as there is tendency
We agree with the reviewer, there are tendency towards statistical significance. The authors consider that tendency is not significant and therefore should not be taken into account.However, a reference to the tendency to statistical significance (p=0.054) for yellowness is reported in '3.1. Chemical Analysis and Physical Properties of the Meat 'section. (Lines 338-339)
Comment 13 Line 246-249: Are your values falling within the recommended range? Please add the recommended range
Range proposed to Shear Force by Marichal et al. (2003) (5.5 to 8.1 kg/cm2) has been included in section '3.1. Chemical Analysis and Physical Properties of the Meat' of goat meat tenderness and compare with other studies. (Lines 319 and 320)
Comment 14 Line 255-257: Did you measure the iron content in the milk they ingested? Do not speculate describe what you did.
We agree with the reviewer. The authors have not measured the iron content in milk because they consider that this food does not include iron. The authors support their discussion in the Argüello et al. (2005) reporting the absence of iron in milk. Several works on the nutritional value of milk indicate that this food is deficient in iron. (Lines 330-333)
Comment 15 Line 259: You did not measure the Orange pulp iron. Furthermore, you incorporated the orange pulp in the diet you did not feed it direct to the animals so you should have measured the iron content of all your diets.
We agree with the reviewer. Unfortunately, the iron content was not measure in mother's diets. During the preparation of the experiment, the authors were informed of the reduced iron content of the orange (0.3mg / 100g fresch). Furthermore, high iron content is not expected in the different diets because they were prepared with the by-products of the orange.
Comment 16 Line 264-269: Please add the range of values of the recommended lightness, redness yellowness, hue angle and chroma etc
We agree with the reviewer. Your contribution clarifies the concept of comparison with other kids goat with similar characteristics. Changes in section '3.1. Chemical Analysis and Physical Properties of the Meat 'have been carried out. (Lines 333-337)
Comment 17 Line 270: This section was poorly done, there is need to explain the mechanism of action of the active ingredients in DOP on fatty acid profiles. DOP has polyphenolic compounds like phenols, vitamin E carotenoids and vitamin C which have antioxidant and antimicrobial properties. The fact that you did not measure them in DOP and your diets makes it discussion of results difficult. You should also have analysed the quantity of vitamin E and total phenols in meat as well this could have made your results easy to discuss.
The authors agree with their suggestion. When designing the experience, the authors considered analyzing the contents of polyphenols, vitamin E carotenoids and vitamin C only in milk. In fact, the results of these analyzes are currently being evaluated with 'major revision' in the journal Animal. However, it was not considered to analyze these compounds in kids meat because kids goat not eaten solid food including vitamin E carotenoids or polyphenols. Furthermore, the animals were slaughtered with just one month of life, so it was unlikely that they had enough antioxidant compounds to influence the oxidation capacity of meat fat. The kids goat in this study have a reduced fat content. In literature, several studies have been carried out to find out the effect antioxidant because of the inclusion of vitamin E in the diet of animals. These works have generally been carried out with animals older than those in the present study. An example in beef is as follows:
Juárez M., Dugan M., Aldai N., Basarab J., and Aalhus J. (2012). Beef quality attributes as affected by increasing the intramuscular levels of vitamin E and omega-3 fatty acids.Meat Science, 90: 764-769.
Comment 18 Line 322-327: Consider deleting these statements.
References to pig in the text were deleted
Comment 19 Line 371-372: Mechanism of action is lacking as well to support your results for example aldehydes are derived from lipid oxidation mallard reaction strecker degradation and breakdown of amino acid through transamination followed by decarboxylation. Hexanal are derived from oxidation of oleic acid whilst benzaldehyde is derived from linoleic acid and also from degradation of phenylalanine.
We agree with the reviewer. Changes in the text in relationship to Mechanism of action have been made about Hexane since it is an important volatile compound in goat. (See 3.3. Volatile Compounds and Aromatic Properties of Meat section) (Lines 450-456)
Comment 20 Line 379-381: Where is Table 6 in this manuscript. Explain the effects of DOP on juiciness and tenderness. There is need to explain the mechanism of action behind this.
It is true, a printing error has hidden Table 6. Table 6 has been included.
Mechanism of action for juiciness and tenderness have been included in section '3.4. Sensorial Properties of Meat' (Lines 476 and 477)
Comment 21 Line 388-390: why?
This is an observation detailed by Martinez-Cerezo et al. (2005). Consumers of ruminant meat from different areas (i.e Northern Europe and Southern Europe) have different tastes. For example, in northern Europe or the British country the consumer values meat with high intensity of flavor from animals slaughtered at advanced ages. However, in southern Europe the consumer prefers light meat, with reduced fat content and reduced flavor intensity. In any case, all products are acceptable in each geographic area. For this reason, the farmer tries to respond to the demand for meat in each geographical area and uses different production strategies.
Comment 22 Line 393-395: Dimethyl disulphide is a volatile sulphur compound which has a significant effect on meat flavour and aroma. This volatile compound is derived from oxidation and degradation of cysteine and methionine sulphur amino acids
We agree. Your contribution improves understanding of the text. In section '3.4. Sensory Properties of Meat 'changes have been made. (Lines 492-494)
Comment 23 Line 402-404: There is a need to explain the mechanism of action
Excuse me, but I don’t understand the question about the mechanism of action required in lines 402-404
The text shown between lines 402-404 refers to an important attribute that the consumers from young ruminant meat usually values. This is the fat content in meat. The amount of fat that the meat presents influences on several attributes such as juiciness, tenderness or flavor. In the kids from present study, the differences in the fat content of the meat of the three treatments are not significant, so large differences in juiciness, tenderness and even the flavor of the meat are not expected between the three groups of animals studied.
Comment 24 Line 433: Overall, the conclusion, needs to be reworked
Conclusions have been rewritten
Comment 25 Line 437-439: Rewrite the conclusion.
Conclusions have been rewritten

Reviewer 4 Report
I reviewed the manuscript entitled "The use of concentrates rich in orange by-products in goat feed and its effects on physico-chemical, textural, fatty acids, volatiles compounds and sensory characteristics of the meat of suckling kids". This study tests the effect on including a by-product form the orange industry in the diet of goats, based on previous study on sheep.
General comments
The manuscript is well written, and the information is well presented. There are some grammar errors in the past tense. The simple form must be used instead of the perfect form; "quality was analysed" instead of "quality has been analysed". There are some points to be improved regarding methodology and presentation of results.
Specific comments
The acronyms must be defined in their first use, e.g. MUFA, PUFA, SFA, FA. Please check throughout the document.
The abstract can be improved by describing the diet and the animal management as well as adding values of the means for the variables reported.
Introduction
L46: the unit for tonnes is "t"
"In fact" is used extensively in the manuscript. In my opinion, most of the times is used unnecessarily. Please check and amend accordingly.
Materials and Methods
L75 calving must be replaced by kidding
L77 please include the mean and SD for liveweight and body condition score for the three groups.
Why did the researchers initially included 46 female goats (it is not even multiple of 3) but then only assessed 10 from each group? This should be explained in the manuscript.
It would help to understand the diets if you first state that "all the diets included x kg/d of alfalfa hay per pregnant goat and xx kg/d per lactating goat". Then specify the kg of concentrate in DO_0.
Is it the dry matter content of the concentrate the same as the citrus pellets? Otherwise the percentage if inclusion of each diet component should be presented in dry matter basis.
I would insert a "_" between DOP and the level of inclusion: DO_0, DO_40 DO_80 to avoid ambiguities.
L79 Insert a comma after "alfalfa hay"
L87-93 this paragraph is chaotic and difficult to follow. I would suggest the following order:
After parturition, kids remained with their dams from birth to weaning with free access to goat milk during all day. All the kids were only fed with maternal milk until slaughtered. Ten male suckling kid goats were selected randomly from each experimental group and slaughtered at an average weight of 8.90 kg (± 0.17) and 32 (±1.5) days old.
The following text should go to Materials and Methods: "During suckling period, two samplings of milk was carried out and representative samples were taken from each dam. Samples were deposited in iceboxes, sent to the laboratory and frozen at -20 °C till analysis.
The following text should go to Results: “The mean values of chemical composition and FA profile of the both mother’s milk sampling are shown in Table 2”.
Please clarify how the two results from each sampling were incorporated into the results (mean of the two samplings?).
Table 2 must be moved to the Results section and a statistical analysis must be performed to assess difference in milk quality regarding feeding groups. This is a crucial analysis since you are assessing the mother's diet on the offspring's meat quality and this is mostly due to milk quality.
Section 2.2 this section should be rephrased: Sampling and Chemical Analysis of Diet and Mother’s Milk
Please explain the procedures to collect feed samples (how, when, how many samples, how much, etc ).
L100-101 you stated the methods used for extract ether and crude fibre and then in L103-105 you describe the method used for "fat" and crude fibre. What is the difference?
L138-140 how many slices per animal? How many pieces per slice?
L141-142 Please provide a reference for this sentence.
L159-160 Please provide a reference for this sentence (FAs).
2.6 Sensorial Evaluation
The experimental design fir this assessment is not clear. You mention "Repeated measures analysis" what usually implies several measurements over the same experimental unit across time. How many times each member of the family ate the same leg? Was it divided in several meals? Please clarify this aspect as it is not clear.
Statistical analysis
Please delete any reference about testing the null hypothesis (equal means) as that is a very basic concept that lies behind almost all the statistical tests.
I would delete from "It is well known..." until "These" as this is well known. The phrase should read " Normality and homoscedasticity assumptions were rejected for many variables according to the xxxxx test"
L193-194 Why did you applied a blanket nonparametric statistical test when the assumptions for ANOVA were met for a few variables (you should state which variables met the assumptions). Given that parametric tests have greater statistical power than non-parametric tests, you might want to keep the ANOVA and improve the changes of finding true effects for the variables that have normal distribution and are homoscedastic.
L203 Please clarify why you mention ANOVA when you used the permutation test.
L204 You stated that you included the individual (animals? test panel members?) in the model but this is not shown in the model included in the following line nor in the description in L205-207. Please clarify.
L211-212 Please clarify the "repeated measures analysis" and the "suggesting performance of a profile analysis". I do not understand what you mean with these statements.
L216 What was the outcome of the normality test? Any test for homoscedasticity?
L217-218 Why not just perform an ANOVA?
Results and Discussion
I would appreciate some discussion of the nutritional value of the three diets, with particular focus on the intermediate value of crude protein for DO_40: why did ii decrease from DO_0 to DO_40and then increased from DO_40 to DO_80? A table with the nutritional composition of each main ingredient of the diet (alfalfa, concentrate and DO) would be helpful to interpret the results.
Table 2 is part of this section and a statistical analysis must be performed on the data to assess differences in milk quality among diet groups.
Could the authors please elaborate a discussion around the effect of fibre content of the diets of lactating goats and their milk fat content? This is relevant since by including increasing levels of orange pellets you are increasing fibre content of the diet which might have an effect on the concentration of milk fat.
L230-231 between brackets you mention a range, but you only report 5.70. A range must have a minimum and a maximum value. Please amend.
L232 I would use "sheep or cattle" (as the animal species) or "lambs and calves" as the livestock category comparable with kids.
L254 "reduced myoglobin content" compared to what? Please elaborate on this.
In some lines such as 285, 287, 290, 319 you mention differences in nutritional composition of the milk to support your findings, but you have not performed any statistical analysis on that database. Therefore, no inference can be done on the effect of milk on meat properties in the current state of this manuscript.
L292 The decrease in MUFA was only true for DO_80. Please rephrase.
L295-296 Please provide a reference to support this statement. Given that both the C18:1 and C18:00 decreased with DO_80... How is it possible that DO favours biohydrogenation? In that case C18:1 should decrease while C18:0 should increase. Please explain.
L301 It seems some word is missing before [30]. What meat property are you referring to?
L311-312 The numerical difference in SFA content among diets was not significant. Therefore, there is no enough evidence to support that SFA content of DO_80 is lower than DO_0 nor DO_40. Statement i L312-313 is not true either Please rephrase L311-313.
L315 Would it be "a desirable n6/n3 ratio? Or a "recommended..."? I think a word is missing there. Please check.
It would ease the interpretation of results in Table 4 if you show the same FAs as in Table 2. Please consider present equivalent tables.
L355 I would put "0.20 and 0.30..." to follow the same order as DO_40 and DO_80.
L371-374 This statement is confusing: "Elmore et al. [42] suggested that a diet high in PUFAs leads to an increase in heptane in grilled lamb meat"... In this case, the diet of the lambs would be the mother's milk. Right? But then you state that "...since the higher heptane content was observed in meat that had a significantly lower content of PUFAs (DOP 80; Table 4)"... Therefore, you are referring to PUFAs in the kind meat AND NOT IN THE KIDS' DIET which is goat's milk. Please amend accordingly so the statements make sense.
SECTION 3.4 CANNNOT BE ASSESSED SINCE THERE IS NO TABLE 6.
L403 This is the only time in the manuscript you refer to a "tendency". However, there were other tendencies according to the tables presented. Please be consistent throughout the manuscript regarding the discussion of non-significant p values.
Figure 1 is the result of a bias in selecting only variables that showed significant effect of the diets. This is not the correct procedure when using a multivariate approach nor the rationale behind its use. Please include all the variables or delete the section and figure as redundant.
Conclusions
After you amend the Resutls and Discussion section pelase make sure your conclussions are still supported by the results.
Author Response
ANSWER TO THE REVIEWER 4
I reviewed the manuscript entitled "The use of concentrates rich in orange by-products in goat feed and its effects on physico-chemical, textural, fatty acids, volatiles compounds and sensory characteristics of the meat of suckling kids". This study tests the effect on including a by-product form the orange industry in the diet of goats, based on previous study on sheep.
General comments
The manuscript is well written, and the information is well presented. There are some grammar errors in the past tense. The simple form must be used instead of the perfect form; "quality was analysed" instead of "quality has been analysed". There are some points to be improved regarding methodology and presentation of results.
Thank you very much for your comments
Specific comments
The acronyms must be defined in their first use, e.g. MUFA, PUFA, SFA, FA. Please check throughout the document.
Acronyms have been revised throughout the document, including Abstract.
The abstract can be improved by describing the diet and the animal management as well as adding values of the means for the variables reported.
Changes in the abstract have been made to describe the diet and the animal management as well as means values for the variables reported.
Introduction
L46: the unit for tonnes is "t"
It is true. ‘tn’ has been changed by ‘t’
"In fact" is used extensively in the manuscript. In my opinion, most of the times is used unnecessarily. Please check and amend accordingly.
The number of 'In fact' expressions currently in the text has been reduced (four times)
Materials and Methods
L75 calving must be replaced by kidding
'Calving' for 'kidding' change has been made (line 85)
L77 please include the mean and SD for liveweight and body condition score for the three groups.
Goat live weight and body condition score for the three groups have been included in section ‘2.1. Animal management and dietary treatments’ (Lines 92 and 93)
Why did the researchers initially included 46 female goats (it is not even multiple of 3) but then only assessed 10 from each group? This should be explained in the manuscript.
This is a misprint. Change in text was performed. 'Previously, 44 first kidding goats were distributed' (Line 85)
Initially 44 goats were available on the experimental ship. There are always a high number of goats since some of them can die. Among them, 10 goats are chosen per group (one kid per calving) because 10 kids were to be analyzed. After reviewing the literature, the authors considered 10 to be an appropriate number for this type of research.
It would help to understand the diets if you first state that "all the diets included x kg/d of alfalfa hay per pregnant goat and xx kg/d per lactating goat". Then specify the kg of concentrate in DO_0.
Details about mother's diet were included in '2.1. Animal management and dietary treatments' section (line 96)
Is it the dry matter content of the concentrate the same as the citrus pellets? Otherwise the percentage if inclusion of each diet component should be presented in dry matter basis.
Table 1 has been modified and units expressed in% dry matter basis
I would insert a "_" between DOP and the level of inclusion: DO_0, DO_40 DO_80 to avoid ambiguities.
We agree with reviewer. This is a good suggestion to facilitate the understanding of the text.
L79 Insert a comma after "alfalfa hay"
Comma was inserted
L87-93 this paragraph is chaotic and difficult to follow. I would suggest the following order:
After parturition, kids remained with their dams from birth to weaning with free access to goat milk during all day. All the kids were only fed with maternal milk until slaughtered. Ten male suckling kid goats were selected randomly from each experimental group and slaughtered at an average weight of 8.90 kg (± 0.17) and 32 (±1.5) days old.
This paragraph was included (Lines 110-113)
The following text should go to Materials and Methods: "During suckling period, two samplings of milk was carried out and representative samples were taken from each dam. Samples were deposited in iceboxes, sent to the laboratory and frozen at -20 °C till analysis.
Thankyou so much. This paragraph was included in ‘2.2. Chemical Analysis of Diet and Mother’s Milk’ section (lines 129-131)
The following text should go to Results: “The mean values of chemical composition and FA profile of the both mother’s milk sampling are shown in Table 2”.
We agree with reviewer. These changes facilitate the understanding of the document. Some paragraphs have been changed place in the text to improve the comprehension of the manuscript . Changes in Table 2 have been included.
Please clarify how the two results from each sampling were incorporated into the results (mean of the two samplings?).
It was clarified in the text as following:
‘The mean values of chemical composition and FA profile of the both mother’s milk from two samplings are shown in Table 2.’ (Lines 279-280)
Table 2 must be moved to the Results section and a statistical analysis must be performed to assess difference in milk quality regarding feeding groups. This is a crucial analysis since you are assessing the mother's diet on the offspring's meat quality and this is mostly due to milk quality.
Table 2 has been moved to the Results section and a statistical analysis was performed.
Section 2.2 this section should be rephrased: Sampling and Chemical Analysis of Diet and Mother’s Milk
Section 2.2 has been rephrased. (line 115)
Please explain the procedures to collect feed samples (how, when, how many samples, how much, etc ).
Details about the procedures to collect feed samples are included in '2.2. Sampling and Chemical Analysis of Diet and Mother’s Milk' section (Lines 116-118)
L100-101 you stated the methods used for extract ether and crude fibre and then in L103-105 you describe the method used for "fat" and crude fibre. What is the difference?
The method references are described on 2.2. Sampling and Chemical Analysis of Diet and Mother’s Milk section (Lines 125-128)
In these lines we describe some details of the analysis carried out to determine each parameter, such as the equipment used.
L138-140 how many slices per animal? How many pieces per slice?
One slice was obtained per animal. Method is detailed in the text (three subsamples by animal) (See 2.3. Slaughter Procedures and Muscle Sampling section) (lines 116-117)
L141-142 Please provide a reference for this sentence.
Reference for this sentence was provided (Guzmán et al., 2019). (line 157)
L159-160 Please provide a reference for this sentence (FAs).
Reference for this sentence was provided (Insausti et al., 2005). (Line 201)
2.6 Sensorial Evaluation
The experimental design fir this assessment is not clear. You mention "Repeated measures analysis" what usually implies several measurements over the same experimental unit across time. How many times each member of the family ate the same leg? Was it divided in several meals? Please clarify this aspect as it is not clear.
The sensorial test was performed as following;
Number of families (X = 10)
- one DOP-0 leg for three members in X family.
- one DOP-40 leg for three members in X family.
- one DOP-80 leg for three members in X family.
- number of legs DOP-0 (n = 10)
- number of legs DOP-40 (n = 10)
- number of legs DOP-80 (n = 10)
Statistical analysis
Please delete any reference about testing the null hypothesis (equal means) as that is a very basic concept that lies behind almost all the statistical tests.
We have maintained these references to clarify what is tested in each analysis.
I would delete from "It is well known..." until "These" as this is well known. The phrase should read " Normality and homoscedasticity assumptions were rejected for many variables according to the xxxxx test"
Text has been corrected (lines 235-237)
L193-194 Why did you applied a blanket nonparametric statistical test when the assumptions for ANOVA were met for a few variables (you should state which variables met the assumptions). Given that parametric tests have greater statistical power than non-parametric tests, you might want to keep the ANOVA and improve the changes of finding true effects for the variables that have normal distribution and are homoscedastic.
At the beginning, for each variable, we ckecked the hypotheses of normality and homoscedasticity. Parametric ANOVA was performed under both hypothesis. A Kruskal.Wallis test was computed when any of these hypothesis were discarded.
However, it was decided to apply the same procedure to all variables to avoid that some variables had more options???? than others to be significant. Anyway, the conclusions tended to agree with?, as we can see in the following tables Column M is 1 when ANOVA parametric test was applied, while M=2 denotes Kruskal-Wallis was applied. Pv is the p- value for the method applied, and finally the p-value for permutation test is in the last column. Note that both p-values tend to agree.
|
0 40 80 M pv pv_perm pH24h 5.932 5.889 6.104 2 0.237 0.219 L. 55.794 54.486 53.646 2 0.150 0.288 a. 4.634 6.648 6.114 2 0.074 0.029 b. 6.724 6.210 7.384 1 0.367 0.347 C. 8.346 9.269 9.673 2 0.146 0.286 H. 54.310 42.840 50.316 2 0.062 0.070 WHC 17.134 16.466 16.152 1 0.619 0.630 Cooking loss 25.756 25.679 24.021 1 0.482 0.492 Mb 2.421 2.228 2.608 1 0.650 0.520 Ash 1.240 1.281 1.342 2 0.606 0.545 Protein 18.683 18.543 18.567 2 0.917 0.981 Fat 1.986 1.830 1.953 1 0.544 0.486 Humidity 77.126 77.482 77.125 2 0.688 0.843 Texture 5.933 5.360 6.557 1 0.067 0.074 |
Fat:
0 40 80 M pv pv_perm
C8.0 0.003 0.005 0.004 1 0.016 0.023
C10.0 0.015 0.013 0.012 1 0.537 0.267
C11.0 0.007 0.006 0.004 2 0.006 0.002
C12.0 0.049 0.050 0.043 2 0.737 0.514
C13.0 0.004 0.004 0.003 2 0.496 0.565
C.8.C13 0.077 0.079 0.067 2 0.368 0.292
C14.0 0.403 0.405 0.357 1 0.506 0.529
C14.1 0.020 0.021 0.017 1 0.478 0.518
C15.0 0.036 0.038 0.035 1 0.665 0.764
C15.1 0.004 0.005 0.004 2 0.150 0.134
C16.0 2.989 2.907 2.697 1 0.332 0.357
C16.1 0.214 0.234 0.187 1 0.148 0.168
C17.0 0.069 0.081 0.078 1 0.035 0.020
C17.1 0.047 0.060 0.054 1 0.059 0.069
C18.0 2.296 2.138 1.864 1 0.000 0.001
C18.1n9t 0.059 0.044 0.043 1 0.015 0.009
C18.1.trans.11..VA. 0.134 0.096 0.081 2 0.001 0.001
C18.1n9c..oleic. 3.641 3.906 3.092 2 0.023 0.018
C18.2n6t 0.025 0.017 0.011 1 0.000 0.001
C18.2n6c..linoleic. 1.567 1.340 1.387 1 0.012 0.013
C18.3.n6.g.linolenic 0.021 0.020 0.018 1 0.071 0.062
C20 0.035 0.034 0.033 1 0.587 0.515
C18.3n3.a.linolenic 0.047 0.051 0.045 1 0.148 0.156
X9c.11t.CLA 0.087 0.081 0.065 1 0.067 0.038
C20.1.n.9 0.041 0.044 0.032 1 0.001 0.001
X10t12c.CLA 0.020 0.012 0.011 2 0.000 0.001
C21 0.015 0.013 0.013 2 0.035 0.139
C20.2 0.211 0.198 0.180 2 0.155 0.142
C22 0.026 0.030 0.023 1 0.001 0.001
C20.3n6 0.066 0.058 0.057 2 0.008 0.009
C22.1.n9.erucic 0.070 0.055 0.056 2 0.002 0.002
C20.4.n6.arachidonic 1.086 1.066 1.074 2 0.940 0.996
C20.3n3 0.012 0.011 0.011 1 0.839 0.873
C23.0 0.011 0.010 0.011 1 0.788 0.728
C20.5n3.EPA. 0.042 0.057 0.050 1 0.009 0.009
C22.2 0.033 0.035 0.028 2 0.001 0.003
C24.0 0.066 0.064 0.053 1 0.002 0.002
C24.1 0.021 0.023 0.019 1 0.345 0.082
C22.5n3..DPA. 0.163 0.162 0.165 1 0.891 0.896
C22.6n3..DHA. 0.066 0.075 0.069 1 0.361 0.245
Volatil compounds:
0 40 80 M pv pv_perm
Acethaldehyde 24375461.6 28444672.50 30766042.80 1 0.173 0.229
Methanethiol 7121190.4 8283273.90 8304084.00 1 0.618 0.659
Ethanol 614302.3 663811.00 795308.80 2 0.481 0.595
X2.Propanone 11154505.1 11235574.70 13129615.10 1 0.280 0.407
Metane.thiobis 14915.0 44371.60 55297.10 2 0.057 0.020
Carbon.disulfide 181362.4 205257.50 185810.60 2 0.907 0.708
X2.Methyl.propanal 7457262.8 8283474.60 8965084.00 2 0.619 0.733
Butanodieno 185592.1 168163.60 190660.80 1 0.889 0.695
X2.Butanona 2835254.8 3173543.60 3959468.10 1 0.235 0.321
X3.Methyl.Butanal 9842382.1 10223394.80 11571731.80 1 0.667 0.746
X2.Methyl.butanal 5626525.2 6316582.20 7405848.70 2 0.494 0.148
Heptano 59714.6 66005.00 159301.20 2 0.018 0.023
Ethyl.furan 145911.4 197830.57 305642.71 2 0.063 NA
Dimethyl.disulfide 1459161.5 1195090.70 2318556.40 1 0.041 0.046
Methyl.benzene 291139.4 289400.33 367924.60 2 0.703 0.684
X3.Methyl.heptano 116585.4 187024.00 63214.71 NA NA NA
n.Nonane 335822.6 334370.33 493871.40 1 0.191 0.465
n.Hexanal 300096.7 348434.11 447412.40 1 0.482 0.724
Heptanal 221371.4 144745.92 138086.50 2 0.058 0.215
X2.3.4.Trisulfide 809918.5 797113.44 762382.10 2 0.509 0.397
Octanal 130138.3 137761.67 136703.00 2 0.671 0.751
Nonanal 48624.8 71252.75 37755.00 NA NA NA
Octanal...nonanal 154450.7 173388.00 144254.00 2 0.727 0.740
L203 Please clarify why you mention ANOVA when you used the permutation test.
ANOVA can be performed using parametric or non-parametric procedures, and within non-parametric procedures, permutation test-based procedures can be used.
L204 You stated that you included the individual (animals? test panel members?) in the model but this is not shown in the model included in the following line nor in the description in L205-207. Please clarify.
Yik is the value for individual k in regime i,
L211-212 Please clarify the "repeated measures analysis" and the "suggesting performance of a profile analysis". I do not understand what you mean with these statements.
This answer has been performed in the text.
‘Two main questions can be addressed for each variable
- Equality of the three mean values.
- If they are not equal, identify the different mean values……..’
L216 What was the outcome of the normality test? Any test for homoscedasticity?
The Mardia test is a generalization of the univariate normality tests based on the skewness and kurtosis measures (Rencher, 2002). We have in this case only one variable, therefore homoscedasticity is not needed to ckeck.
L217-218 Why not just perform an ANOVA?
Taking into account that each familiy is observed in each regime, the Independence of observations is not acceptable. Profile analysis overrides this limitations.
Results and Discussion
I would appreciate some discussion of the nutritional value of the three diets, with particular focus on the intermediate value of crude protein for DO_40: why did ii decrease from DO_0 to DO_40and then increased from DO_40 to DO_80? A table with the nutritional composition of each main ingredient of the diet (alfalfa, concentrate and DO) would be helpful to interpret the results.
Excuse me. I don’t exactly understand your suggestion. The mothers' diets were isoenergetic and isoproteic, as detailed in the text and in Table 1.
Table 2 is part of this section and a statistical analysis must be performed on the data to assess differences in milk quality among diet groups.
Tabla 2 was included in ‘Results and discussion section’. This table includes a statistical analysis.
Could the authors please elaborate a discussion around the effect of fibre content of the diets of lactating goats and their milk fat content? This is relevant since by including increasing levels of orange pellets you are increasing fibre content of the diet which might have an effect on the concentration of milk fat.
Comments about the effect of fiber have been included in the text. See '3. Results and discussion' section. (Lines 290-292)
L230-231 between brackets you mention a range, but you only report 5.70. A range must have a minimum and a maximum value. Please amend.
This is true. We agree with reviewer. Sentence 'pH values in range 5.70' was changed by sentence 'pH values around 5.70' (Line 302)
L232 I would use "sheep or cattle" (as the animal species) or "lambs and calves" as the livestock category comparable with kids.
We agree with reviewer. we are changed 'calves' by 'cattle' in '3.1. Chemical Analysis and Physical Properties of the Meat' section (Line 303)
L254 "reduced myoglobin content" compared to what? Please elaborate on this.
The authors do not intend to compare myoglobin content among animals. However, reference for comparison with adult animals has been reported in the text.(Lines 326 and 327)
In some lines such as 285, 287, 290, 319 you mention differences in nutritional composition of the milk to support your findings, but you have not performed any statistical analysis on that database. Therefore, no inference can be done on the effect of milk on meat properties in the current state of this manuscript.
We agree with reviewer. That is why a new table 2 with statistical comparison of the composition of milk has been included in the manuscript.
L292 The decrease in MUFA was only true for DO_80. Please rephrase.
A new redaction was included in the manuscript. 'Although mother's milk did not show significant changes in the content of the main unsaturated FA (C18:1) when orange pulp was included in the goats' diet (Table 2), a significant decrease in MUFA content was observed in meat from suckling kids from DOP-80 (Table 4)' (Lines 362-365)
L295-296 Please provide a reference to support this statement. Given that both the C18:1 and C18:00 decreased with DO_80... How is it possible that DO favours biohydrogenation? In that case C18:1 should decrease while C18:0 should increase. Please explain.
Thanks for your suggestion. It is very valuable guidance. The authors had made an interpretation mistake. A new text in the manuscript has been proposed. The fat biohydrogenation in suckling kids are reduced and do not explain the biohydrogenation of fatty acids. In suckling kids, the fatty acid profile of meat is mainly related to the fatty acid profile of ingested milk. (Lines 369-371)
L301 It seems some word is missing before [30]. What meat property are you referring to?
That's true. 'flavour meat' was included in the text (Lines 377 and others)
L311-312 The numerical difference in SFA content among diets was not significant. Therefore, there is no enough evidence to support that SFA content of DO_80 is lower than DO_0 nor DO_40. Statement i L312-313 is not true either Please rephrase L311-313.
We agree with reviewer. Sentence was rephrased en '3.2. Fatty acids and Nutritional Properties of the Meat' section. According numerical difference in C18:0 (Table 4), SFA was changed by C18:0. Stearic fatty acid is an important saturated fatty acid in meat. (Lines 386-388)
L315 Would it be "a desirable n6/n3 ratio? Or a "recommended..."? I think a word is missing there. Please check.
This is true. We agree with reviewer. Also, Wood et al. (2004) reported 'recommended'. Then..... 'recommended n6 / n3 ratio' ...was included in the text. (Line 390)
It would ease the interpretation of results in Table 4 if you show the same FAs as in Table 2. Please consider present equivalent tables.
Equivalences in tables 2 and 4 have been made to facilitate the interpretation of the results.
L355 I would put "0.20 and 0.30..." to follow the same order as DO_40 and DO_80.
Change of order has been made to facilitate the understanding of the text
L371-374 This statement is confusing: "Elmore et al. [42] suggested that a diet high in PUFAs leads to an increase in heptane in grilled lamb meat"... In this case, the diet of the lambs would be the mother's milk. Right? But then you state that "...since the higher heptane content was observed in meat that had a significantly lower content of PUFAs (DOP 80; Table 4)"... Therefore, you are referring to PUFAs in the kind meat AND NOT IN THE KIDS' DIET which is goat's milk. Please amend accordingly so the statements make sense.
We agree with reviewer. A new sentence has been included in '3.3. Volatile Compounds and Aromatic Properties of Meat' section (Linees 460-464)
SECTION 3.4 CANNNOT BE ASSESSED SINCE THERE IS NO TABLE 6.
Excuse me. A printing mistake has occurred. Table 6 was included
L403 This is the only time in the manuscript you refer to a "tendency". However, there were other tendencies according to the tables presented. Please be consistent throughout the manuscript regarding the discussion of non-significant p values.
We agree. The authors have removed the reference to the 'tendency' of the Overall appraisal of kid's meat since the effect was not significant (p = 0.07) (Table 6).
Figure 1 is the result of a bias in selecting only variables that showed significant effect of the diets. This is not the correct procedure when using a multivariate approach nor the rationale behind its use. Please include all the variables or delete the section and figure as redundant.
Thank you so much for your suggestion.
The authors consider this is a common practice in presenting results with a large number of variables studied. We must take into account that the number of variables studied is much higher than the number of animals analyzed.
The discriminant analysis has been presented with the variables that were significant since the authors consider the following:
We have quantifyed the contribution of each of the variables in the significant differences observed between the three treatments. This is not possible with ANOVA analysis.
It is intended to show a graphic separation of the effects that the inclusion of PDO in the ration of the mothers has on the quality of the meat of the kids.
The aim is to simplify the model for its understanding. In this way, the variables that could are coincident were eliminated.
For this reason, we do not consider that the discriminant analysis is redundant, but rather provides information to understand the results obtained.
If these arguments are not robust, the authors could remove the section.
Conclusions
After you amend the Resutls and Discussion section pelase make sure your conclussions are still supported by the results.
Thanks for your comments. The conclusions of the manuscript have been validated and rewritten taking into account the arguments collected in the results and the discussion.

Reviewer 5 Report
The submitted manuscript reports data on the use of concentrates rich in orange by-products in goat feed and its effects on physico-chemical, textural, fatty acids, volatiles compounds and sensory characteristics of the meat of suckling kids. The topic is interesting and falls within the scope of the journal. Despite the authors presented a wide range of parameters, there are scientific limitations in the manuscript. The overall presentation of manuscript needs some improvements. The abstract lacks specific results to highlight the key findings and some digestive data for kids. Apart from these main arguments, the manuscript would have benefited from careful editing before submission as language and formatting are below the standard.
Specific comments:
Line 81: Ingredients in Table 1 should be presented as format on % air dry matter or dry matter, but not as % fresh matter basis.
Line 87: the sentence that….. kid goats were selected randomly from each experimental group and slaughtered at an average weight of 8.90 kg (± 0.17) and 32 (±1.5) days old disaccord with the statement of bodyweight that the average slaughter live and cold kid carcass weights were 8.98, 9.03 and 8.65 kg and 4.74, 4.82 and 4.65 kg for DOP 0, DOP 40 and DOP 80, respectively (Line 223). Please provide with data of initial weight, final weight, bodyweight gain and dietary DM intake of animals in a Table.
Author Response
ANSWER TO THE REVIEWER 5
Line 81: Ingredients in Table 1 should be presented as format on % air dry matter or dry matter, but not as % fresh matter basis.
This is a misprint, the data was actually expressed in dry matter basis. For this reason, Table 1 has been corrected.
Line 87: the sentence that….. kid goats were selected randomly from each experimental group and slaughtered at an average weight of 8.90 kg (± 0.17) and 32 (±1.5) days old disaccord with the statement of bodyweight that the average slaughter live and cold kid carcass weights were 8.98, 9.03 and 8.65 kg and 4.74, 4.82 and 4.65 kg for DOP 0, DOP 40 and DOP 80, respectively (Line 223). Please provide with data of initial weight, final weight, bodyweight gain and dietary DM intake of animals in a Table.
Reference to the slaughter live weight of the line 87 Animals (8.90 kg) is in agreement with the average slaughter weight reference of animals in the three treatments (8.98 + 9.03 + 8.65) / 3 = 8.88kg). This value refers to the commercial slaughter weight of suckling animals. Values of 4.74, 4.82 and 4.65 kg in line 223 are in reference to the cold carcass weight of the animals.
The birth weight at the three groups has been added in the text (at the beginning of ‘3. Results and discussion’): “The average birth, slaughter live and cold kid carcass weights were 2.84, 3.06 and 2.95 kg, 8.98, 9.03 and 8.65 kg and 4.74, 4.82 and 4.65 kg for DOP- 0, DOP-40 and DOP-80, respectively. There were no significant differences between the treatments in terms of birth weight, slaughter weight and cold carcass weight (p>0.05)”.
We do not consider adding a table with all the data on Growth performance of kids since these are included in another article (Using dried orange pulp in the diet of dairy goats: effects on milk yield and composition and blood parameters of dams and growth performance and carcass quality of kids), currently in ‘major revision’ in Animal journal.
The ingestion of milk DM could not be taken into account due to difficulties in measuring this value. The kids drank milk from the mother directly.

Round 2
Reviewer 1 Report
I have given a quick view to the authors responses and to the new proposal of the article.
In the response letter of authors, I have checked that if the authors supported some of their ideas in other authors like Aldai (2006) for FA analysis, they are not citing the primary author but a secondary (that is an author of this paper). On the other hand, they have incorrectly cited one of my articles (Bravo-Lamas et al. 2018) to support that a FA analysis with an identification of 37FA is enoughly good, while the aim of using GC columns of 100 or 120 meters with a proper method in ruminant tissues is to avoid the overlapping of FAs and therefore, to avoid the incorrect interpretations of data. Thus, a careful FA analysis can give more than 100 FAs in ruminant tissues (Bravo-Lamas et al, 2016, 2018a, 2018b), both in milk and meat. These incorrect arguments related to FA analysis make me think that the rest of the analysis is not thorough like it should be. It is common when in one paper there are many analyses done in low number of samples and that was one of my reasons for rejecting this paper.
On the other hand, although I see that the authors have worked in the improvement of the paper, in my quick revision of the paper I realised that they are comparing the FA of milk with FA of meat when the data are given in different units: mg/100g DM of milk and mg/100g fresh meat. That is not correct. When the tissue has a high percentage of water (that is the case of milk, and even of meat), data are magnified when is given by 100g DM, so both data should be in the same units, by 100g of fresh milk and meat, or by 100g of DM, or by % of total FA. That makes me think again that authors are not acting carefully. The same happended with the discussion, where I missed some references to support their hypothesis.
On the other hand, when they explain issues about volatile compounds and sensory analysis (in the responses document and in the paper) the authors explain that volatile compounds and therefore sensory characteristics are born in meat when it is cooked, that is correct. However, these volatiles are born by the simultaneous reactions of several compounds, including FAs, proteins and other minority components like vitamins or minerals. So, although it could be a good idea trying to find a relationship between FA and volatile pattern, taking into account the inclusion of DOP in the goats feed, it should be even more important to analyze with detail milk characteristics (chemical, FA and volatile compounds) in order to see how it can affect meat characteristics (before and after of cooked).
These with other reasons I already concluded during my first revision carry me to suggest the rejection of this article and to these authors to make a thorough analysis of less parameters of milk of goat, and therefore, to study the consequences in suckling kids.
Hope you find my letters useful.
Author Response
ANSWER TO THE REVIEWER 1 (rev 2)
I have given a quick view to the authors responses and to the new proposal of the article.
We appreciate your work for manuscript improvement.
In the response letter of authors, I have checked that if the authors supported some of their ideas in other authors like Aldai (2006) for FA analysis, they are not citing the primary author but a secondary (that is an author of this paper). On the other hand, they have incorrectly cited one of my articles (Bravo-Lamas et al. 2018) to support that a FA analysis with an identification of 37FA is enoughly good, while the aim of using GC columns of 100 or 120 meters with a proper method in ruminant tissues is to avoid the overlapping of FAs and therefore, to avoid the incorrect interpretations of data. Thus, a careful FA analysis can give more than 100 FAs in ruminant tissues (Bravo-Lamas et al, 2016, 2018a, 2018b), both in milk and meat. These incorrect arguments related to FA analysis make me think that the rest of the analysis is not thorough like it should be. It is common when in one paper there are many analyses done in low number of samples and that was one of my reasons for rejecting this paper.
Thank you for your helpful comments. We have revised our paper accordingly and feel that your comments helped clarify and improve our paper
The authors consider that there are written documents that recommend the use of SUPELCO 37 for determining the basic profile of fatty acids in meat. The document that has been presented is not intended to make a more precise analysis of the different fatty acids in meat (that is, cis, trans isomers etc). We are trying to have enough information (representatives or conventional fatty acids) to have consistent idea about the effect on meat quality. The focus is avoiding differences when producers introduce by products in the animal diet. We understand the limitations of analyzing only 37 fatty acids and present some of the most relevant fatty acids for the interpretation of the results. We understand your point of view and thank very much your contribution
Reference to Aldai et al (2006) were not included because the proposed method had some adaptations to the method that have been collected in Horcada et al. in several manuscripts.
On the other hand, although I see that the authors have worked in the improvement of the paper, in my quick revision of the paper I realised that they are comparing the FA of milk with FA of meat when the data are given in different units: mg/100g DM of milk and mg/100g fresh meat. That is not correct. When the tissue has a high percentage of water (that is the case of milk, and even of meat), data are magnified when is given by 100g DM, so both data should be in the same units, by 100g of fresh milk and meat, or by 100g of DM, or by % of total FA. That makes me think again that authors are not acting carefully. The same happended with the discussion, where I missed some references to support their hypothesis.
We understand your point of view, but we consider that the different units used to show the fatty acid profile of milk and meat is not limiting for the discussion of the results. We have not applied statistical analysis but we have tried to find relationship between both results.
The authors presented the results shown in Table 2 (milk analysis) to describe the composition of milk in relation to DM. This method is considered valid since milk has a high amount of water. This value is always higher than the water content of the meat. Frequently, in the literature, the fatty acid profile of milk is presented in relation to DM. The authors have attempted to characterize the composition of milk based on DM and to be able to refer to other manuscripts on this matter. Likewise, in characterizing the fatty acid profile of meat (Table 4), the authors considered that the units referring to fresh meat were correct, since there are many manuscripts in the literature. In this sense, presenting the units according to fresh meat allows us to have a real idea of the nutritional value of the meat. This idea is presented in the work since ratios related to the health of consumers are shown.
In either case, dry matter or fresh matter, the percentage of fatty acids with respect to the total fatty acids detected remains constant.
On the other hand, when they explain issues about volatile compounds and sensory analysis (in the responses document and in the paper) the authors explain that volatile compounds and therefore sensory characteristics are born in meat when it is cooked, that is correct. However, these volatiles are born by the simultaneous reactions of several compounds, including FAs, proteins and other minority components like vitamins or minerals. So, although it could be a good idea trying to find a relationship between FA and volatile pattern, taking into account the inclusion of DOP in the goats feed, it should be even more important to analyze with detail milk characteristics (chemical, FA and volatile compounds) in order to see how it can affect meat characteristics (before and after of cooked).
We agree with Reviewer and we will consider your valorous comment in futures studies
At work, much more can be done to test the effect of including the pulp of the orange in the goat's diet on the characteristics of the meat of the kids. In fact, another paper has been submitted for review and is currently under 'major review' in another journal. This work fundamentally analyzes the aspects related to the characteristics of milk and its antioxidant capacity.
However, analyzing the characteristics of the meat before and after cooking was not the experimental reason for the design of the work. Reviewer 1 must understand that a project establishes hypotheses prior to design and the results must satisfy that hypothesis. The idea of analyzing changes in the composition of volatile meat compounds before and after cooking was not considered in the work that has been submitted for evaluation. The aim is to characterize goat meat from three feeding models of its mothers. In this sense, the limitation of samples to analyze must also be taken into account. The amount of reference muscle in the case of kids is very limited and the idea of the manuscript is to analyze various meat quality parameters (proximal composition, texture, color, volatiles, lipid profile and sensory analysis).
We will take into account your suggestion for a new project that specifically analyzes the influence of the cooking method of meat on the composition of volatile compounds in kid meat.
These with other reasons I already concluded during my first revision carry me to suggest the rejection of this article and to these authors to make a thorough analysis of less parameters of milk of goat, and therefore, to study the consequences in suckling kids.
Hope you find my letters useful.
Reviewer 4 Report
The manuscript has improved comapred with the submitted version. However, there are still some aspects that require clarification:
L94 "Changes in feed composition did not influence the palatability of the diets"... How did you test that?
Table 1 Crude protein DOP-0: 17.76, DOP-40:16.59, DOP-80: 18.43; DIP: 10.5, 11.0, 11.5… Are the diets isoproteic?
L117 "equal amounts of three subsamples collected during the suckling period"… when? If I want to repeat the experiment what should I do?
L247 "the random effect of the individual"… you did not include the random effect of the individual. You are referring to the experimental error, which is always part of the model.
L252 "The goats’ commercial traits"… what are theoe traits?
L252-253 "The goats’ commercial traits, ME, and FA composition for milk during the suckling period were analysed with the repeated measures procedure"… but in Table 2 you state that is the mean value of two samples. Where is the effect of the week of lactation?
L258-259 Could you please add the statistical model for the repeated measures design?
L270-271 "Finally, a linear discriminant analysis was conducted with all the variables to discriminate between the three feeding regimes" and L504 "all the present datasets of suckling kid goat meat"... But then in L5140-515 you state “only those variables that were significantly different (p < 0.1) according to a permutation ANOVA test were selected in the model” … Oo, is it all the data set or only the significant variables? Are significant at 0.05 or at 0.10?
L338-339 "A close and significant trend was observed for yellowness (p = 0.054) was observed (Table3)" … What is a significant trend? What is the range you use for the p values to state that there is a trend? There are other p values between 0.05 and 0.10 that were not considered as traits. Again, please be consistent regarding discussing no significant values.
Author Response
ANSWER TO THE REVIEWER 4 (rev 2)
The manuscript has improved comapred with the submitted version. However, there are still some aspects that require clarification:
Thank you for your helpful comments. We have revised our paper accordingly and feel that your comments helped clarify and improve our paper
L94 "Changes in feed composition did not influence the palatability of the diets"... How did you test that?
Goats were subjected to a long period of adaptation to DOP consumption, so that, the palatability of DOP would logically not affect the intake of each of the experimental groups during the development of the experience. From the beginning of the third month of gestation, DOP was gradually introduced into the diet of the animals in the DOP-40 and DOP-80 groups. Effectively, before application the experimental diets, visual observation of feed intake indicated that goats consumed all DOP pellets offered. In addition and although the individual intake was not measured, the average intake per animal obtained indirectly through the consumption of the group shows similar values in the different dietary treatments (DM was 1.85, 1.82 and 1.80 kg/d, protein digestible in the intestine was 0.19, 0.20 and 0.20 kg/d, and gross energy was 8.42, 8.10 and 8.00 Mcal/d in the DOP-0, DOP-40 and DOP-80 groups, respectively). Therefore, we have considered it appropriate to include additional information in M&M to better explain this aspect (lines 94-97 and 103-107).
Table 1 Crude protein DOP-0: 17.76, DOP-40:16.59, DOP-80: 18.43; DIP: 10.5, 11.0, 11.5… Are the diets isoproteic?
The CP in ruminants is not suitable for a correct protein assessment of the ration, the really important parameter in ruminants is the PDI; in this case, there was only a maximum difference percentage of one point between the three diets. Although the % PDI have not been exactly the same, because it is very difficult to adjust the ration with these ingredients, these small differences seem not to affect the average intake in the different groups. In addition, milk production and proximal chemical composition were not affected in early lactation, results that are currently being reviewed in other work.
L117 "equal amounts of three subsamples collected during the suckling period"… when? If I want to repeat the experiment what should I do?
As suggested, information on this aspect has been added (lines 122-123).
L247 "the random effect of the individual"… you did not include the random effect of the individual. You are referring to the experimental error, which is always part of the model.
We refer to the experimental error, but this a random variate which is defined for each individual.
L252 "The goats’ commercial traits"… what are theoe traits?
The "commercial traits" are frequently used in relation to the content of dry matter, crude protein and fat milk (see Table 2). However, the authors agree to change by "proximal chemical composition" (Line 255)
L252-253 "The goats’ commercial traits, ME, and FA composition for milk during the suckling period were analysed with the repeated measures procedure"… but in Table 2 you state that is the mean value of two samples. Where is the effect of the week of lactation?
We consider that the factor week of lactation is not relevant for this work and for simplification, the results for this factor have not been presented in this paper. Therefore, we have considered it appropriate to include a sentence in M&M to better explain this aspect (lines 258-265).
L258-259 Could you please add the statistical model for the repeated measures design?
Statistical model for the repeated measure desings was included (lines 274-277)
L270-271 "Finally, a linear discriminant analysis was conducted with all the variables to discriminate between the three feeding regimes" and L504 "all the present datasets of suckling kid goat meat"... But then in L5140-515 you state “only those variables that were significantly different (p < 0.1) according to a permutation ANOVA test were selected in the model” … Oo, is it all the data set or only the significant variables? Are significant at 0.05 or at 0.10?
We agree,
Lines 270 and 271 were changed by "variables that were significantly different" (lines 282-283)
Line 504 was changed: A linear discriminant analysis model was built to determine the relationship between groups of variables and the three feeding groups for datasets that were significantly different of suckling kid goat meat (Figure 1). (line 513)
Line 514. We agree with the reviewer. There are many opinions in the literature about p-values between 0.05 and 0.1, the most accepted is to say that it is a weakly significant p-value, but in this case we named 0.05 p-value (line 523)
L338-339 "A close and significant trend was observed for yellowness (p = 0.054) was observed (Table3)" … What is a significant trend? What is the range you use for the p values to state that there is a trend? There are other p values between 0.05 and 0.10 that were not considered as traits. Again, please be consistent regarding discussing no significant values.
We agree with reviewer.
Authors had considered trend p = 0.054 and p = 0.084 to discuss the results. However, with in order to be consistent in the discussion, these references have been removed.
Reference to trend in yelowness of meat and C18:1 in mother's milk were deleted.
Text was rewrited (lines 370-376)

Reviewer 5 Report
I have check the revised version of Manuscript ID: animals-754966 and I believe the manuscript has been significantly improved and now warrants publication in Animals.
Author Response
ANSWER TO THE REVIEWER 5 (Rev 2)
I have check the revised version of Manuscript ID: animals-754966 and I believe the manuscript has been significantly improved and now warrants publication in Animals.
Dear Reviewer 5,
thank you very much for your work. The revision of manuscripts is a work that contributes to the dissemination of scientific results. The disinterested work of researchers is very important for the maintenance of scientific publications.
